# PPM1H phosphatase counteracts LRRK2 signaling by selectively dephosphorylating Rab proteins

Kerryn Berndsen[1†], Pawel Lis[1†], Wondwossen M Yeshaw[2], Paulina S Wawro[2], Raja S Nirujogi[1], Melanie Wightman[1], Thomas Macartney[1], Mark Dorward[1], Axel Knebel[1], Francesca Tonelli[1], Suzanne R Pfeffer[2], Dario R Alessi[1*]

[1]MRC Protein Phosphorylation and Ubiquitylation Unit, School of Life Sciences, University of Dundee, Dundee, United Kingdom; [2]Department of Biochemistry, Stanford University School of Medicine, Stanford, United States

**Abstract** Mutations that activate LRRK2 protein kinase cause Parkinson's disease. LRRK2 phosphorylates a subset of Rab GTPases within their Switch-II motif controlling interaction with effectors. An siRNA screen of all human protein phosphatases revealed that a poorly studied protein phosphatase, PPM1H, counteracts LRRK2 signaling by specifically dephosphorylating Rab proteins. PPM1H knockout increased endogenous Rab phosphorylation and inhibited Rab dephosphorylation in human A549 cells. Overexpression of PPM1H suppressed LRRK2-mediated Rab phosphorylation. PPM1H also efficiently and directly dephosphorylated Rab8A in biochemical studies. A "substrate-trapping" PPM1H mutant (Asp288Ala) binds with high affinity to endogenous, LRRK2-phosphorylated Rab proteins, thereby blocking dephosphorylation seen upon addition of LRRK2 inhibitors. PPM1H is localized to the Golgi and its knockdown suppresses primary cilia formation, similar to pathogenic LRRK2. Thus, PPM1H acts as a key modulator of LRRK2 signaling by controlling dephosphorylation of Rab proteins. PPM1H activity enhancers could offer a new therapeutic approach to prevent or treat Parkinson's disease.
DOI: https://doi.org/10.7554/eLife.50416.001

*For correspondence:
d.r.alessi@dundee.ac.uk

[†]These authors contributed equally to this work

Competing interests: The authors declare that no competing interests exist.

## Introduction

LRRK2 is a large multidomain protein enzyme consisting of a Roc-Cor family GTPase followed by a serine/threonine kinase domain (*Cookson, 2010*; *Gilsbach and Kortholt, 2014*). Missense mutations within the GTPase domain (N1437H, R1441G/C, Y1699C) and kinase domains (G2019S, I2020T) hyper-activate LRRK2 protein kinase and cause Parkinson's disease (PD) (*Zimprich et al., 2004*; *Paisán-Ruíz et al., 2004*; *Alessi and Sammler, 2018*). Mutations in LRRK2 are one of the most common genetic causes of familial Parkinson's comprising ~5% of familial Parkinson's, and ~1% of sporadic Parkinson's patients (*Healy et al., 2008*; *Domingo and Klein, 2018*). In terms of clinical presentation and late age of onset, LRRK2 mediated Parkinson's closely resembles the common sporadic form of the disease affecting the vast majority of patients (*Marras et al., 2011*). The G2019S mutation, located within the magnesium binding motif of the kinase domain, is the most common LRRK2 mutation and activates LRRK2 kinase activity around 2-fold (*West et al., 2005*; *Greggio et al., 2006*; *Jaleel et al., 2007*). Penetrance of the G2019S and other pathogenic LRRK2 mutations is incomplete (*Ruiz-Martínez et al., 2010*; *Goldwurm et al., 2011*). There is emerging evidence that the LRRK2 pathway is hyperactivated in some patients with idiopathic PD (*Di Maio et al., 2018*). Pharmaceutical companies have developed LRRK2 inhibitors for treatment and prevention of PD (*Alessi and Sammler, 2018*) and clinical trials have commenced and/or are planned (see https://clinicaltrials.gov/ NCT03976349 (BIIB094) and NCT03710707 (DNL201)).

Recent studies have revealed that LRRK2 phosphorylates a subgroup of Rab proteins (Rab3A/B/C/D, Rab8A/B, Rab10, Rab12, Rab29, Rab35, and Rab43) at a conserved Thr/Ser residue (Thr73 for Rab10), located at the center of the effector binding switch-II motif (*Steger et al., 2016*; *Steger et al., 2017*; *Jeong et al., 2018*). Consistent with Rab proteins comprising disease-relevant substrates, all established pathogenic mutations enhance LRRK2 mediated Rab protein phosphorylation in a manner that is blocked by diverse LRRK2 inhibitors (*Steger et al., 2016*; *Steger et al., 2017*; *Ito et al., 2016*). LRRK2 phosphorylation of Rab proteins blocks the ability of Rab proteins to interact with cognate effectors such as GDI and guanine nucleotide exchange factors, thereby trapping the phosphorylated Rab protein in the GTP bound state on the membrane where it has been phosphorylated (*Steger et al., 2016*; *Pfeffer, 2018*). Recent work identified a novel group of effectors including RILPL1 and RILPL2 that bind preferentially to LRRK2 phosphorylated Rab8 and Rab10 (*Steger et al., 2017*). LRRK2 phosphorylated Rab8A and Rab10, in complex with RILPL1/RILPL2, inhibit the formation of primary cilia that are implicated in controlling a Sonic hedgehog-driven neuroprotective pathway that could provide a mechanism by which LRRK2 is linked to PD (*Dhekne et al., 2018*). Other research has revealed that components implicated in PD including Rab29 (*Purlyte et al., 2018*; *Liu et al., 2018*) and VPS35 (*Mir et al., 2018*) also regulate phosphorylation of Rab proteins via LRRK2.

The protein phosphatase(s) that act on LRRK2 phosphorylated Rab proteins have not been identified, but appear to be highly active, as treatment of cell lines with LRRK2 inhibitors induce rapid dephosphorylation of Rab10 within 1-2 min (*Ito et al., 2016*). In contrast, dephosphorylation of LRRK2 at a well-studied potential autophosphorylation site (Ser935) occurs much more slowly in response to LRRK2 inhibitor administration, requiring up to 90 min (*Ito et al., 2016*). Only a small proportion (around 1%) of the total cellular Rab proteins are phosphorylated (*Steger et al., 2016*; *Ito et al., 2016*). In this study, we set out to identify and characterize the protein phosphatase(s) that counteract LRRK2 signaling by dephosphorylating LRRK2-modified Rab proteins.

## Results

### PP1 or PP2A phosphatases do not dephosphorylate Rab proteins

We first explored whether PP1 (Protein Phosphatase-1) or PP2A (Protein Phosphatase-2A) previously implicated in regulating LRRK2 (*Lobbestael et al., 2013*; *Athanasopoulos et al., 2016*), control Rab phosphorylation. Wild-type and LRRK2[R1441C] knock-in mouse embryonic fibroblasts (MEF cells) were treated with 0.1 μM calyculin-A (PP1 inhibitor) or 1 μM Okadaic acid (PP2A inhibitor) (*Ishihara et al., 1989*), and neither of these agents had a significant impact on basal phosphorylation of Rab10 (Thr73) or Rab12 (Ser105) or dephosphorylation following administration of the well-characterized highly selective and potent LRRK2 inhibitor termed MLi-2 (*Fell et al., 2015*) (*Figure 1*). Consistent with PP1 phosphatase regulating dephosphorylation of LRRK2 at Ser935 (*Lobbestael et al., 2013*), calyculin-A stimulated Ser935 phosphorylation, particularly in the LRRK2[R1441C] MEF cells. It also prevented dephosphorylation of Ser935 following administration of MLi-2 (Figure 1). At 1 μM, Okadaic acid is likely to partially inhibit PP1 (*Ishihara et al., 1989*); this concentration also blocked MLi-2 from inducing dephosphorylation of Ser935. Calyculin-A, but not okadaic acid promoted phosphorylation of the PP1 substrate MYPT1 (Thr853) (*Dimopoulos et al., 2007*; *Hutchinson et al., 2011*; *Westphal et al., 1999*). Calyculin-A and okadaic acid also activated the S6K1 kinase pathway as judged by increase in S6K1 phosphorylation (Thr389, mTORC1 site) that is regulated by PP1 and PP2A (*Figure 1*).

### siRNA screen to identify phosphatases that regulate Rab10 phosphorylation

To identify protein phosphatases that act on LRRK2-phosphorylated Rab10, we undertook a phosphatase focused siRNA screen in human A549 adenocarcinomic alveolar basal epithelial cells that express endogenous LRRK2 (*Lis et al., 2018*). We assembled a library consisting of 322 siRNAs (Dharmacon) targeting 264 phosphatase catalytic subunits (189 protein phosphatases, 26 halo acid dehalogenase phosphatases, 18 chloroperoxidases, 5 nucleoside-diphosphate-linked moiety-X phosphatases and 7 carbohydrate phosphatases) as well as 56 characterized regulatory subunits (*Supplementary file 1*). Two replicate screens were undertaken in which cells were treated with

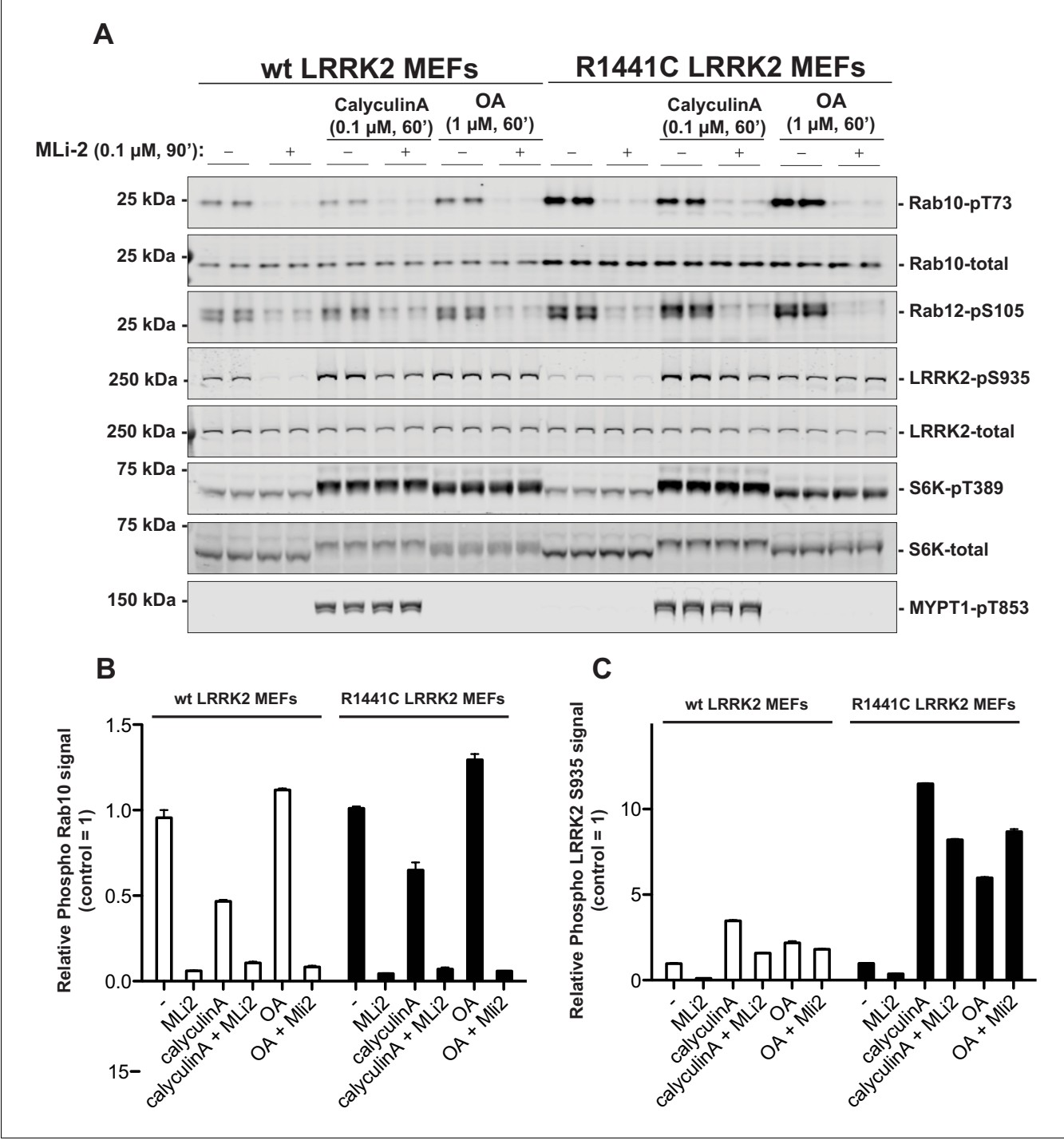

**Figure 1.** PP1 or PP2A phosphatases do not dephosphorylate Rab proteins. Littermate-matched wild-type and R1441C LRRK2 MEF cells were pre-treated with or without 100 nM MLi-2 for 90 min before administration of either 100 nM Calyculin A or 1 µM Okadaic Acid for further 60 min. Floating and adhering cells were lysed, and 10 µg of whole cell extract subjected to quantitative immunoblot analysis with the indicated antibodies (all at 1 µg/ml). Each lane represents cell extract obtained from a different dish of cells (two replicates per condition). (A) The membranes were developed using the LI-COR Odyssey CLx Western Blot imaging system. (B and C) Immunoblots were quantified for phospho-Thr73 Rab10/total Rab10 ratio (B) and phospho-Ser935 LRRK2/total LRRK2 ratio (C) using the Image Studio software. Data are presented relative to the phosphorylation ratio observed in cells treated with no MLi-2 and no phosphatase inhibitor, as mean ± SD.

DOI: https://doi.org/10.7554/eLife.50416.002

siRNA targeting each phosphatase component for 72 h before cell lysis (Screen 1 and 2, *Figure 2A and B*). A third screen was also performed in which partial dephosphorylation of Rab10 was induced by adding MLi-2 inhibitor for 5 min prior to cell lysis (Screen 3, *Figure 2A and C*). As a readout of LRRK2 kinase activity we monitored phosphorylation of endogenous Rab10 employing a previously developed specific phospho-antibody that robustly detects endogenously LRRK2 phosphorylated Rab10 by immunoblot analysis in whole cell extracts (*Lis et al., 2018*). Due to the availability of this antibody and the relatively high abundance of Rab10 in most cells, monitoring Thr73 Rab10

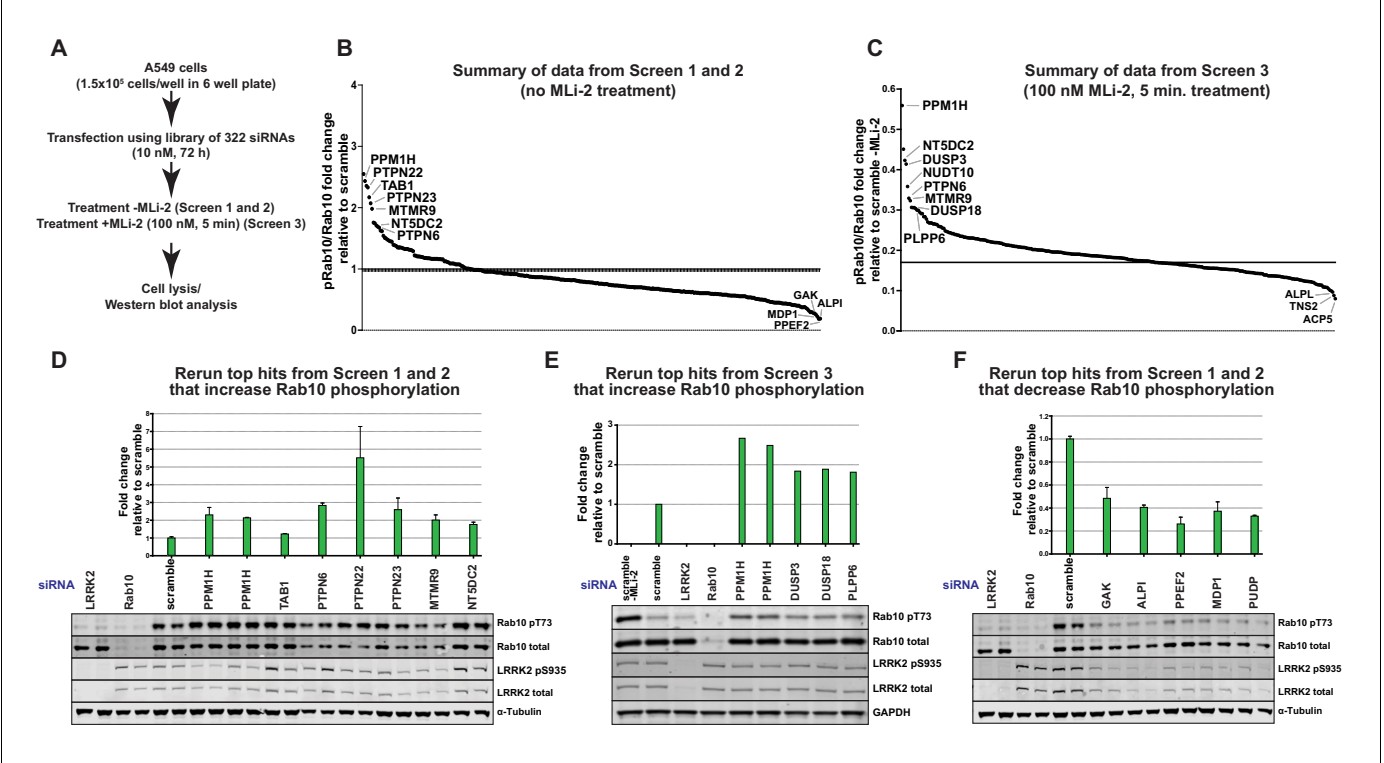

**Figure 2.** siRNA screens to identify phosphatases that regulate Rab10 phosphorylation. (**A**) Workflow of siRNA knockdown screens employed in this study. Human A549 cells were transfected with siRNA pools (Dharmacon) for 72 hr targeting 264 phosphatase catalytic subunits (189 protein phosphatases, 26 halo acid dehalogenase phosphatases, 18 chloroperoxidases, five nucleoside-diphosphate-linked moiety-X phosphatases and seven carbohydrate phosphatases) as well as 56 characterized regulatory subunits. The list of phosphatase components targeted, and oligonucleotide sequences utilized is provided in *Supplementary file 1*. Cells were lysed and immunoblotted for total LRRK2, LRRK2 pS935, total Rab10, and Rab10 pT73 and immunoblots developed using the LI-COR Odyssey CLx Western Blot imaging system. The ratio of phospho-Thr73 Rab10/total Rab10 in each sample was quantified using the Image Studio software. (**B**) Summary of results from Screen 1 and 2 (without MLi-2 pretreatment). Calculated ratio of phosphorylated Rab10 and total Rab10 relative to the scrambled siRNA control, ranked from highest increase in Rab10 phosphorylation to the strongest decrease (mean of the two replicates, SD values not shown on the chart, numerical data provided in *Supplementary file 1*). (**C**) As in (**B**), summary of results from Screen 3 (in which cells were treated for 5 min with 100 nM MLi-2 treatment prior to cell lysis), ratio calculated relative to scrambled siRNA control without MLi-2 treatment. (**D**) Human A549 cells were transfected with siRNA for 72 hr with the indicated top hits from screen 1 and 2 that increased Rab10 phosphorylation. Cells were then lysed and immunoblotted with the indicated antibodies, analyzed as described above. Data are presented relative to the ratio of Rab10 phosphorylation/total Rab10 observed in cells treated with scrambled siRNA as mean ± SD. (**E**) As in (**D**) except the key hits that increased Rab10 phosphorylation in Screen three were reanalyzed. Cells were also treated with 100 nM MLi-2 prior to cell lysis. (**F**) As in (**D**) except the key hits that decreased Rab10 phosphorylation in Screen 1 to 3 were reanalyzed.

DOI: https://doi.org/10.7554/eLife.50416.003

The following figure supplements are available for figure 2:

**Figure supplement 1.** Primary data from siRNA Screen 1 Human A549 cells were transfected with the indicated siRNA pools (Dharmacon) for 72 hr.
DOI: https://doi.org/10.7554/eLife.50416.004

**Figure supplement 2.** Primary data from siRNA Screen 2 Human A549 cells were transfected with the indicated siRNA pools (Dharmacon) for 72 hr.
DOI: https://doi.org/10.7554/eLife.50416.005

**Figure supplement 3.** Primary data from siRNA Screen 3 Human A549 cells were transfected with the indicated siRNA pools (Dharmacon) for 72 hr.
DOI: https://doi.org/10.7554/eLife.50416.006

phosphorylation is currently the most facile readout of LRRK2 kinase activity. Phospho-Thr73 Rab10 and total Rab10 were quantified in parallel employing a multiplex immunoblot assay. The ratio of phospho-Thr73 Rab10 versus total Rab10 was used to calculate the degree of Rab10 phosphorylation in each cell lysate. Levels of total LRRK2 and Ser935 phosphorylated LRRK2 were also analyszed. The results are summarized in *Figure 2B* (Screen 1 and 2) and *Figure 2C* (Screen 3). The primary immunoblotting data for each lysate is presented in *Figure 2—figure supplement 1* (Screen 1), *Figure 2—figure supplement 2* (Screen 2) and *Figure 2—figure supplement 3* (Screen 3).

The top hits that enhanced Rab10 protein phosphorylation over 1.5-fold in Screen 1 and 2 were PPM1H (Metal-dependent protein phosphatases [PPM] family serine/threonine phosphatase), TAB1 (PPM family catalytically inactive pseudophosphatase), PTPN6 (tyrosine phosphatase) PTPN22 (tyrosine phosphatase), PTPN23 (tyrosine phosphatase), MTMR9 (myotubularin lipid phosphatase) and NT5DC2 (nucleoside-diphosphate-linked moiety-X phosphatase) (*Figure 2B*). In screen 3 the top hits were PPM1H, DUSP3 (dual specificity phosphatase), DUSP18 (dual specificity phosphatase) and PLPP6 (presqualene diphosphate phosphatase) (*Figure 2C*). We also identified 5 phosphatases (GAK, ALPI, PPEF2, MDP1, PUDP) that when knocked-down, significantly reduced Rab10 phosphorylation (*Figure 2B*).

## Further analysis of siRNA screen phosphatase hits

PPM1H was observed as a significant hit in all 3 screens. It was also the top serine/threonine phosphatase in each of the screens. Human PPM1H possesses 514 residues encompassing a PPM type phosphatase domain with no other known functional motifs (*Chen et al., 2017*). Repeat studies in multiple experiments confirmed that siRNA mediated depletion of PPM1H increased Rab10 phosphorylation, without affecting overall levels of Rab10 or LRRK2 (*Figures 2D, E* and *3A*). Using a newly generated sheep polyclonal antibody, we established that PPM1H expression was significantly lowered by the siRNA treatment (*Figure 3A*). Consistent with previous work (*Lee-Hoeflich et al., 2011*), we observed that endogenous PPM1H, detected by immunoblot, migrates as a doublet of ~55 kDa (reason for doublet unknown) with the top band migrating at the expected molecular weight for full length PPM1H. Expression of both forms of PPM1H was lowered by siRNA treatment. We also confirmed that siRNA knockdown of NT5DC2 (*Figure 2D*), DUSP3, DUSP18 and PLPP6 (*Figure 2E*) enhanced Rab10 phosphorylation without impacting total Rab10 or LRRK2 levels, or LRRK2 Ser935 phosphorylation. However, knock-down of the tyrosine phosphatases PTPN6, PTPN22, PTPN23 and the possible inositol phosphatase MTMR9 significantly reduced Rab10 protein levels (without affecting total LRRK2) and therefore potentially increase the calculated phospho-Rab10/total Rab10 ratio (*Figure 2D*). siRNA depletion of TAB1 enhanced Rab10 protein phosphorylation to a marginal extent in repeat experiments (*Figure 2D*). Knock-down of GAK, ALPI, PPEF2, MDP1, and PUDP markedly decreased LRRK2 expression, likely accounting for the decreases in Rab10 phosphorylation observed (Figure 2F). These phosphatases were not investigated further in this study.

## Overexpression of PPM1H inhibits LRRK2-mediated Rab protein phosphorylation

Overexpression of wild-type, PPM1H strikingly ablated the phosphorylation of Rab10 observed following overexpression of the LRRK2[R1441G] pathogenic mutant (*Figure 3B*). To generate a catalytically inactive mutant of PPM1H we mutated the essential catalytic His153 that is conserved in all PPM1H phosphatases (*Chen et al., 2017*). We found that mutation of His153 to residues such as Leu that was used in a previous study (*Lee-Hoeflich et al., 2011*), markedly reduced protein expression, whereas mutation of this residue to Asp preserved similar expression to wild-type levels. Overexpression of PPM1H[H153D] catalytic mutant failed to induce dephosphorylation of Rab10 (*Figure 3B*) Similarly, overexpression of wild-type, but not catalytically inactive PPM1H blocked phosphorylation of Rab8A (*Figure 3C*) as well as Rab3A, Rab3B, Rab3C, Rab3D, Rab8B, Rab12, Rab35 and Rab43 (*Figure 3—figure supplement 1A–1H*). In contrast, overexpression of PPM1H did not suppress phosphorylation of Rab8A at the Ser111 site induced following activation of the PINK1 kinase by addition of the mitochondrial uncoupler, CCCP (*Lai et al., 2015*) (*Figure 3—figure supplement 2A*). In addition, overexpression of PPM1H did not impact the phosphorylation of key markers of the Akt and AMPK signaling pathways (*Figure 3—figure supplement 2B*).

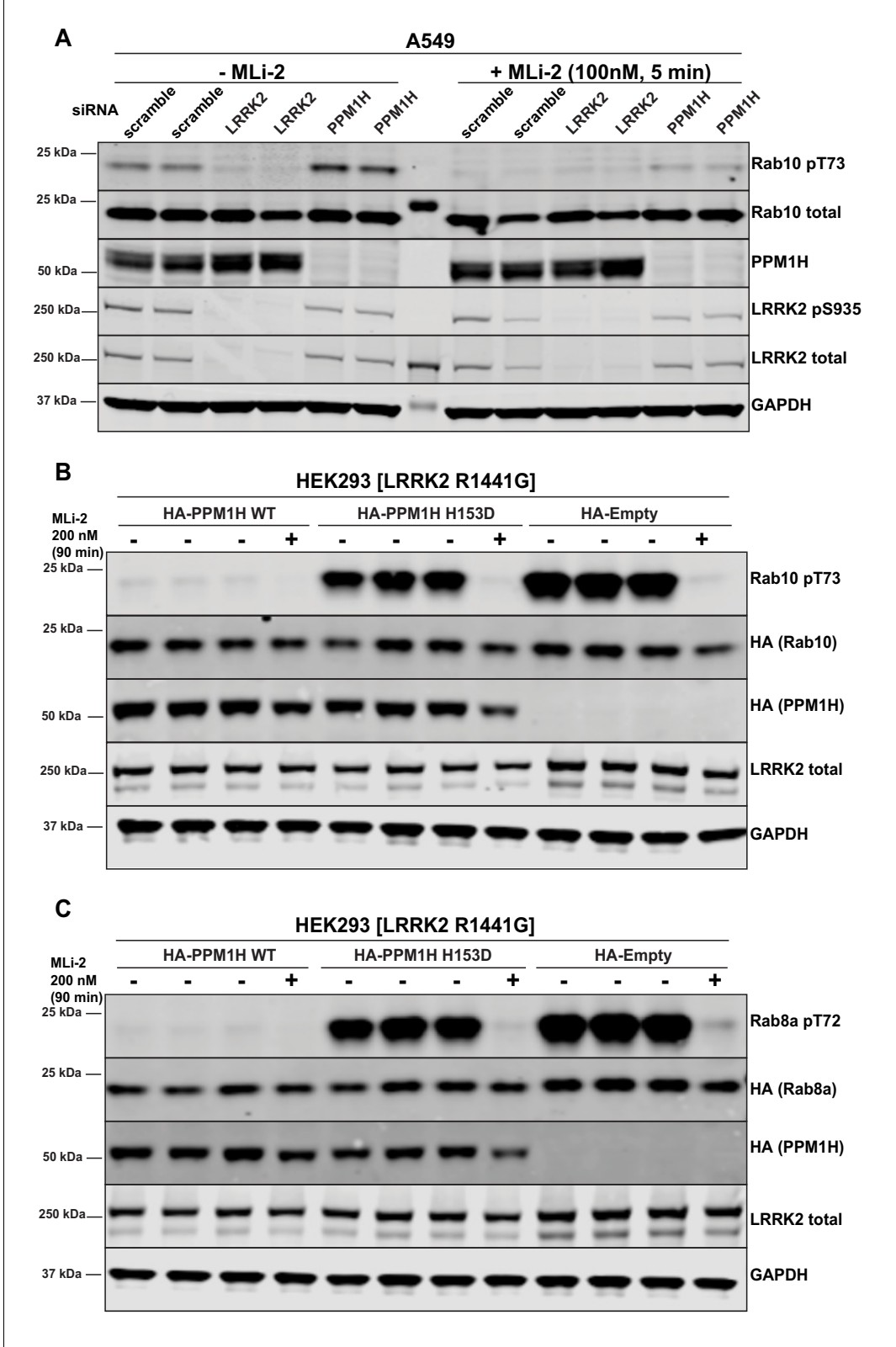

**Figure 3.** Further analysis of siRNA screen phosphatase hits. (**A**) Human A549 cells were transfected with the indicated siRNA pools (Dharmacon) for 72 hr. Cells were treated as indicated ±100 nM MLi-2 for 5 min prior to lysis and 10 μg of whole cell extract subjected to quantitative LI-COR immunoblot analysis with the indicated antibodies (all at 1 μg/ml). Each lane represents cell extract obtained from a different dish of cells (two replicates per condition). (**B and C**) HEK293 cells were transiently transfected with constructs expressing the indicated components. 24 hr post-transfection, cells were

*Figure 3 continued on next page*

*Figure 3 continued*

treated with ±200 nM MLi-2 for 90 min and then lysed. 10 µg whole cell lysate was subjected to immunoblot analysis with the indicated antibodies at 1 µg/mL concentration and membranes analyzed as in (A). Each lane represents cell extract obtained from a different dish of cells (three replicates per condition without MLi-2 treatment, one replicate per condition with MLi-2 treatment).

DOI: https://doi.org/10.7554/eLife.50416.007

The following figure supplements are available for figure 3:

**Figure supplement 1.** PPM1H dephosphorylates multiple LRRK2 phosphorylated Rab proteins.

DOI: https://doi.org/10.7554/eLife.50416.008

**Figure supplement 2.** PPM1H does not dephosphorylate Ser111 of Rab8A or key phosphorylation sites of AMPK and Akt signaling pathways.

DOI: https://doi.org/10.7554/eLife.50416.009

We also studied LRRK2-mediated phosphorylation of GFP-Rab10 stably expressed in wild-type MEF cells, upon overexpression of wild-type or catalytically inactive forms of PPM1H, by immunofluorescence microscopy (*Figure 4*). As observed previously, in cells not expressing either PPM1H form, phosphorylated Rab10 (detected using a phosphoRab10-specific antibody) was concentrated over peri-centriolar membranes, with a narrower localization than total Rab10 in these cells (*Dhekne et al., 2018*). However, in cells expressing wild-type PPM1H (but not its catalytically inactive forms), levels of phosphorylated Rab10 were markedly reduced if not abolished (*Figure 4*).

We next tested the two phosphatases most closely related to PPM1H termed PPM1J and PPM1M (*Chen et al., 2017*). We found that PPM1M but not PPM1J suppressed Rab10 and Rab8A phosphorylation in cells, albeit to a lower extent than PPM1H (*Figure 5A*). There are 20 PPM family

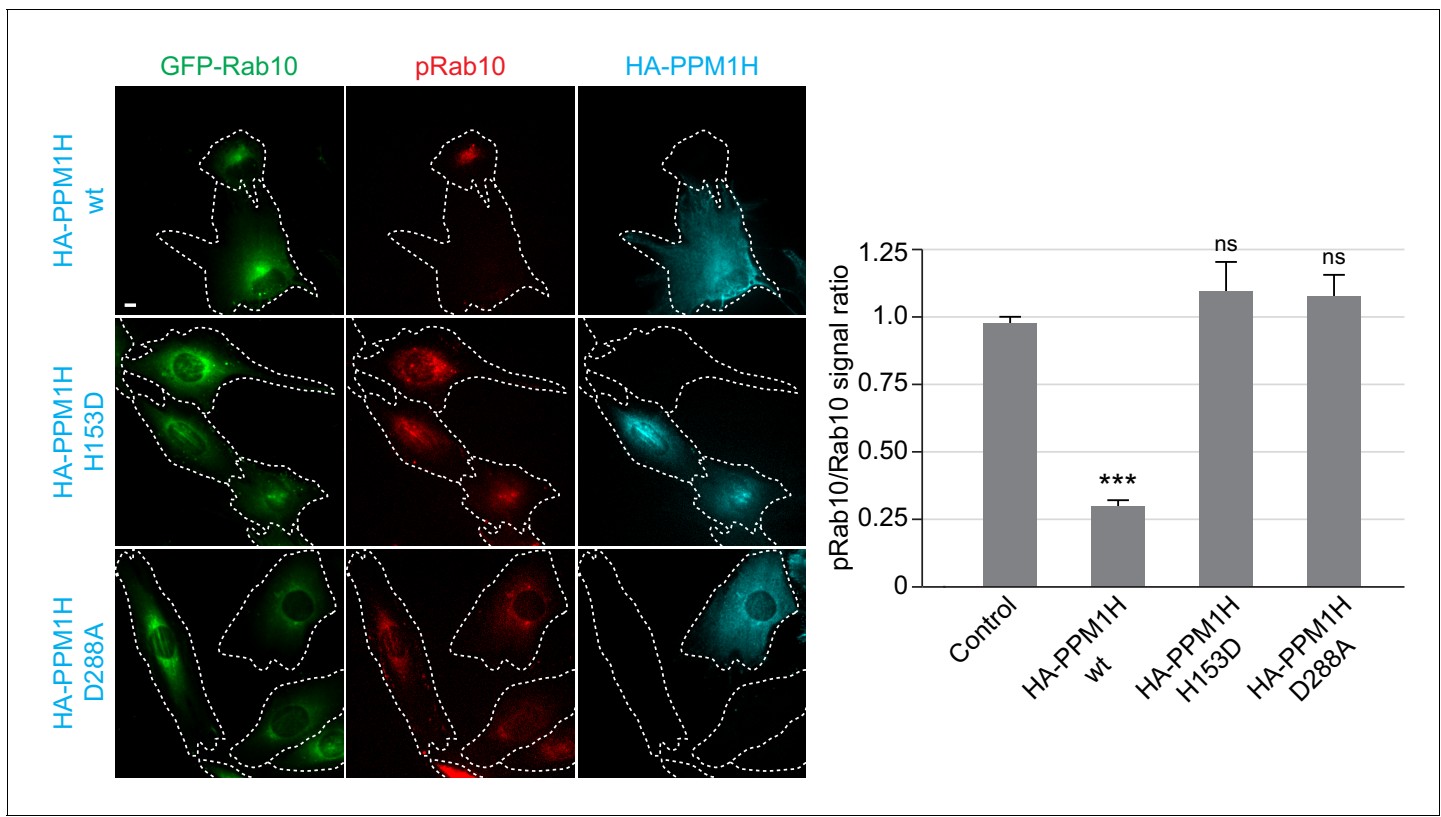

**Figure 4.** HA-PPM1H expression decreases phospho-Rab10 levels in MEF cells. Wild-type MEF cells stably expressing GFP-Rab10 (green) were transiently transfected with plasmids encoding HA-PPM1H wild-type, HA-PPM1H H153D, or HA-PPM1H D288A as indicated at left. After 24 hr the cells were subsequently fixed and stained with mouse anti-HA antibody (shown in blue) and rabbit anti-phospho-Rab10 antibody (shown in red). Scale bar, 10 µm. Data were quantified by determining the ratio of phospho-Rab10 to GFP-Rab10 signal and normalizing these values to the ratio in cells not expressing PPM1H. For each condition, at least 20 cells were analyzed. ***, p<0.001.

DOI: https://doi.org/10.7554/eLife.50416.010

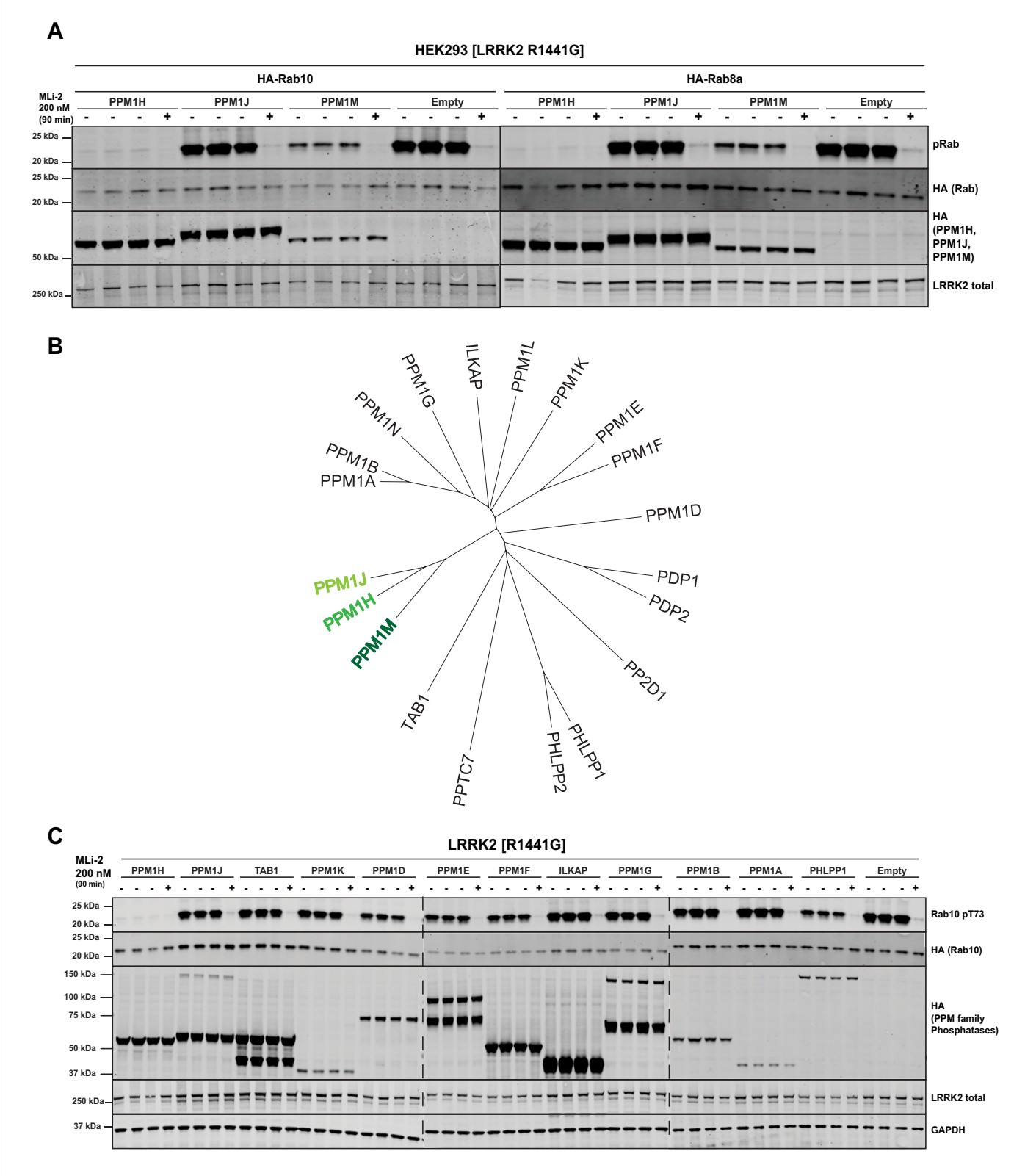

**Figure 5.** PPM1M, but not other PPM family members tested dephosphorylate Rab10. (**A**) HEK293 cells were transiently transfected with the indicated constructs including wild-type PPM1H subfamily phosphatases (PPM1H, PPM1M and PPM1J). 24 hr post-transfection, cells were treated with ±200 nM MLi-2 for 90 min and then lysed. 10 μg whole cell lysate was subjected to immunoblot analysis with the indicated antibodies at 1 μg/mL concentration and membranes analyzed using the OdysseyClx Western Blot imaging system. Each lane represents cell extract obtained from a different dish of cells
*Figure 5 continued on next page*

*Figure 5 continued*

(three replicates per condition without MLi-2 treatment, one replicate per condition with MLi-2 treatment). (B) Phylogenetic tree of the 20 PPM family members generated using the MEGA7 software (*Kumar et al., 2016*). (C) As In (A) except that HEK293 cells were transiently transfected with constructs expressing the indicated PPM family members.

DOI: https://doi.org/10.7554/eLife.50416.011

The following figure supplement is available for figure 5:

**Figure supplement 1.** HEK293 cells were transiently transfected with the indicated constructs.

DOI: https://doi.org/10.7554/eLife.50416.012

phosphatases in humans (*Figure 5B*). In addition to PPM1M and PPM1J, we probed 10 other PPM members (TAB1, PPM1K, PPM1D, PPM1E, PPM1F, ILKAP, PPM1G, PPM1B, PPM1A, PHLPP1) and found that only PHLPP1 (*Brognard et al., 2007*) moderately reduced Rab10 phosphorylation (*Figure 5C*). Overexpression of other phosphatases identified in the siRNA screen had no (MTMR9, PTPN23) or only moderate effects (PLPP6, NT5DC2) on Rab10 phosphorylation (*Figure 5—figure supplement 1*).

## CRISPR knock-out of PPM1H enhances Rab10 protein phosphorylation

We next deployed a CRISPR-Cas9 gene editing approach to knock-out PPM1H in A549 cells. As mentioned above, immunoblotting studies suggested that there are potentially at least two forms of PPM1H expressed in cells. CRISPR-Cas9 guides were designed to target exon-1 of the PPM1H gene and would be predicted to ablate expression of the full length PPM1H protein but may not impact expression of a shorter splice variant that lacks exon-1. We isolated 5 independent clones that demonstrated complete loss of the full length PPM1H protein by immunoblot analysis. These clones still possessed the lower species detected with the PPM1H antibody and indeed, expression of this form appeared to be upregulated in two of the cell lines (*Figure 6A*). We also generated a complete PPM1H knock-out cell line in which both the upper and lower species of PPM1H were knocked-out, using CRISPR-Cas9 guides targeting exon 4 of PPM1H. Basal levels of Rab10 phosphorylation was increased 2-5-fold in 10 independent knock-out cell lines that we examined compared to 4 wild-type cell clones that had been through all of the equivalent steps including clonal selection (*Figure 6A*).

To assess the stoichiometry of Rab10 phosphorylation in cell extracts derived from wild-type and PPM1H knock-out cell lines, we utilized a parallel reaction monitoring mass spectrometry approach. Cell extracts were subjected to electrophoresis on a polyacrylamide gel and the region of the gel containing Rab proteins excised, digested with trypsin and spiked with known amounts of heavy labeled tryptic peptides corresponding to dephosphorylated and phosphorylated forms of Rab10. Following mass spectrometry, we assessed absolute levels of phospho- and dephospho-Rab10 peptides encompassing Thr73 by referencing levels to the internal heavy labeled peptide standards. This revealed that the stoichiometry of Rab protein phosphorylation in wild-type A549 cells was ~0.8% (*Figure 6B*), similar to values obtained in previous work in different cell types using mass spectrometry (*Steger et al., 2016*) or Phos-tag analysis (*Ito et al., 2016*). In the PPM1H knock-out cells the stoichiometry of Rab protein phosphorylation increased to ~2.6% (*Figure 6B*).

## Evidence that PPM1H regulates dephosphorylation of Rab8A and Rab35 in addition to Rab10

To assess whether PPM1H knock-out results in increased phosphorylation of endogenous Rab proteins other than Rab10, we performed an immunoprecipitation experiment using a pan- phospho-specific-Rab antibody that cross-reacts with LRRK2 phosphorylated Rab3, Rab8, Rab10, Rab35 and Rab43 (*Lis et al., 2018*). This approach has previously been exploited to demonstrate that these Rab proteins are phosphorylated by LRRK2 in vivo (*Steger et al., 2017*). Immunoblot analysis of the pan-phospho-specific-Rab immunoprecipitates revealed that in addition to Rab10, phosphorylated forms of Rab8A and Rab35 were enriched in the PPM1H knock-out cell lines in comparison to wild-type (*Figure 6C*). We were unable to detect Rab3A or Rab43 in these studies, suggesting that the levels of the LRRK2 phosphorylated forms of these substrates are low in A549 cells. Immunoblotting of cell extracts with a recently developed phospho-Rab12 antibody that specifically detects LRRK2 phosphorylated Rab12 reveals that LRRK2 phosphorylated Rab12 levels were similar in wild-type and

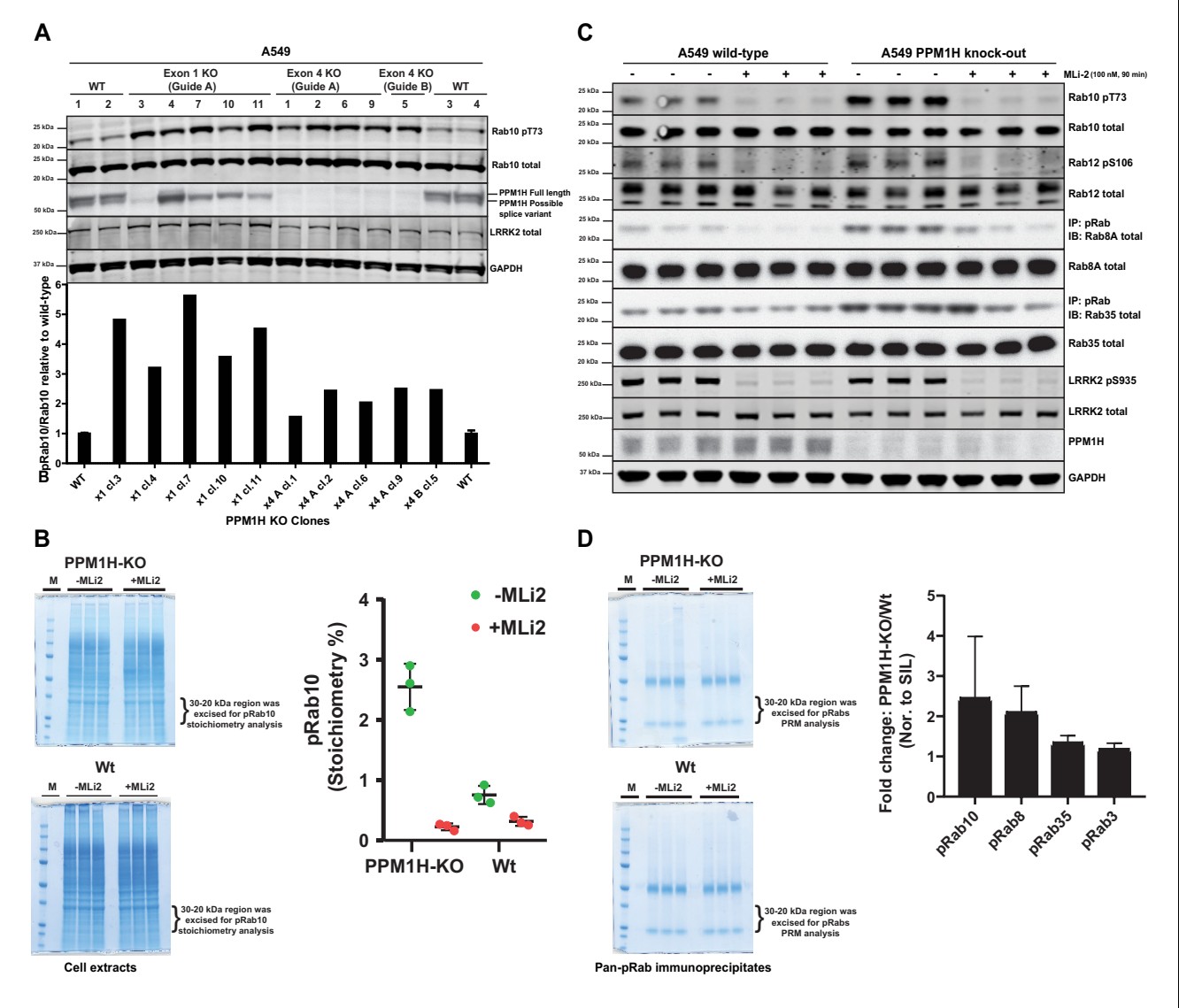

**Figure 6.** CRISPR knock-out of full length PPM1H isoform enhances Rab protein phosphorylation. (**A**) Four independent wild-type (WT 1, WT 2, WT 3, WT 4) and ten independent CRISPR/Cas9 PPM1H knock-out A549 cells (using three different sets of guide RNAs) that had all been through in parallel, single-cell sorting and expansion procedures were analyzed. 20 μg whole cell lysate was subjected to immunoblot analysis with the indicated antibodies at 1 μg/mL concentration and membranes analyzed using the OdysseyClx Western Blot imaging system. The ratio of phospho-Thr73 Rab10/ total Rab10 was quantified using Image Studio software and data presented relative to the phosphorylation ratio observed in wild-type cells. (**B**) 50 μg of cell lysate from three wild-type and three PPM1H knock-out clones, treated ±MLi-2 (100 nM, 90 min) was resolved on a 4–12% Bis-Tris gradient gel. Following colloidal Coomassie staining, the lane encompassing 30 to 20 kDa region was excised and subjected to in-gel digestion using trypsin. The digest was spiked with an equimolar ratio of 10 fmol heavy synthetic phosphorylated Rab10 Thr73 (FHpTITTSYYR) and non-phosphorylated Rab10 Thr73 (FHTITTSYYR) peptides and analyzed on a QE-HFX mass spectrometer in PRM mode. The raw data were processed using Skyline software suite and the light/heavy ratios for both phosphorylated and non-phosphorylated Thr73 peptides were used in determining the pRab10 (FHpTITTSYYR) stoichiometry. The stoichiometry of pRab10 for each sample is indicated and error bars represent the mean with SD (n = 3 in each condition). (**C**) Immunoprecipitation using the pan-phospho-specific pRab8 antibody was performed using 2 mg cell lysate from three wild-type and three PPM1H knock-out clones, treated - ± MLi-2 (100 nM, 90 min). Immunoprecipitates were analyzed by immunoblotting with indicated antibodies. Rab10 pT73 and Rab12 pS106 phosphorylation was assessed in whole cell lysates. (**D**) The pRab8 antibody immunoprecipitates from PPM1H-knock-out±MLi2 and wild-type ±MLi2 was resolved on a 4–12% Bis-Tris gradient gel. Following colloidal Coomassie staining, the lane encompassing the 30 to 20 kDa region was excised and subjected to In-gel digestion using trypsin and spiked with an equimolar ratio of 50 fmol heavy synthetic pRab10, pRab8, pRab3, pRab35, pRab1 and pRab43 phosphopeptides along with 25 fmol of 'Pierce Peptide Retention Time Calibration Mixture' peptides. Sample were analyzed on QE-HFX mass spectrometer using PRM mode. The raw data were processed using Skyline software suite and the light/heavy ratios of phosphorylated

*Figure 6 continued on next page*

*Figure 6 continued*

Rab proteins were determined; the fold change between wild-type and knock-out cells is plotted with error bars representing the mean ± SD (n = 3 in each condition). The raw data is included in *Supplementary file 2*.

DOI: https://doi.org/10.7554/eLife.50416.013

PPM1H knock-out cell extracts (*Figure 6C*). In addition, we subjected the pan-specific phospho-Rab immunoprecipitates to mass spectrometry analysis, which confirmed significantly higher levels of Rab8 and Rab10, and to a lesser extent Rab35, in samples immunoprecipitated from wild-type cells (*Figure 6D*). We were unable to accurately quantify levels of other Rab isoforms, as the level of detection of these was too low.

## PPM1H knock-out results in delayed Rab10 dephosphorylation in response to MLi-2 inhibitor treatment

We next undertook a time course of dephosphorylation of phospho-Rab10 in response to administration of MLi-2 LRRK2 inhibitor in wild-type and PPM1H knock-out cells. In the wild-type cells MLi-2 induced rapid dephosphorylation of Rab10 with ~50% dephosphorylation observed after 1 min, ~75% dephosphorylation after 2 min with only basal levels observed by 5 min. In the PPM1H knock-out cells, dephosphorylation of PPM1H was approximately 2-fold slower, with ~30%

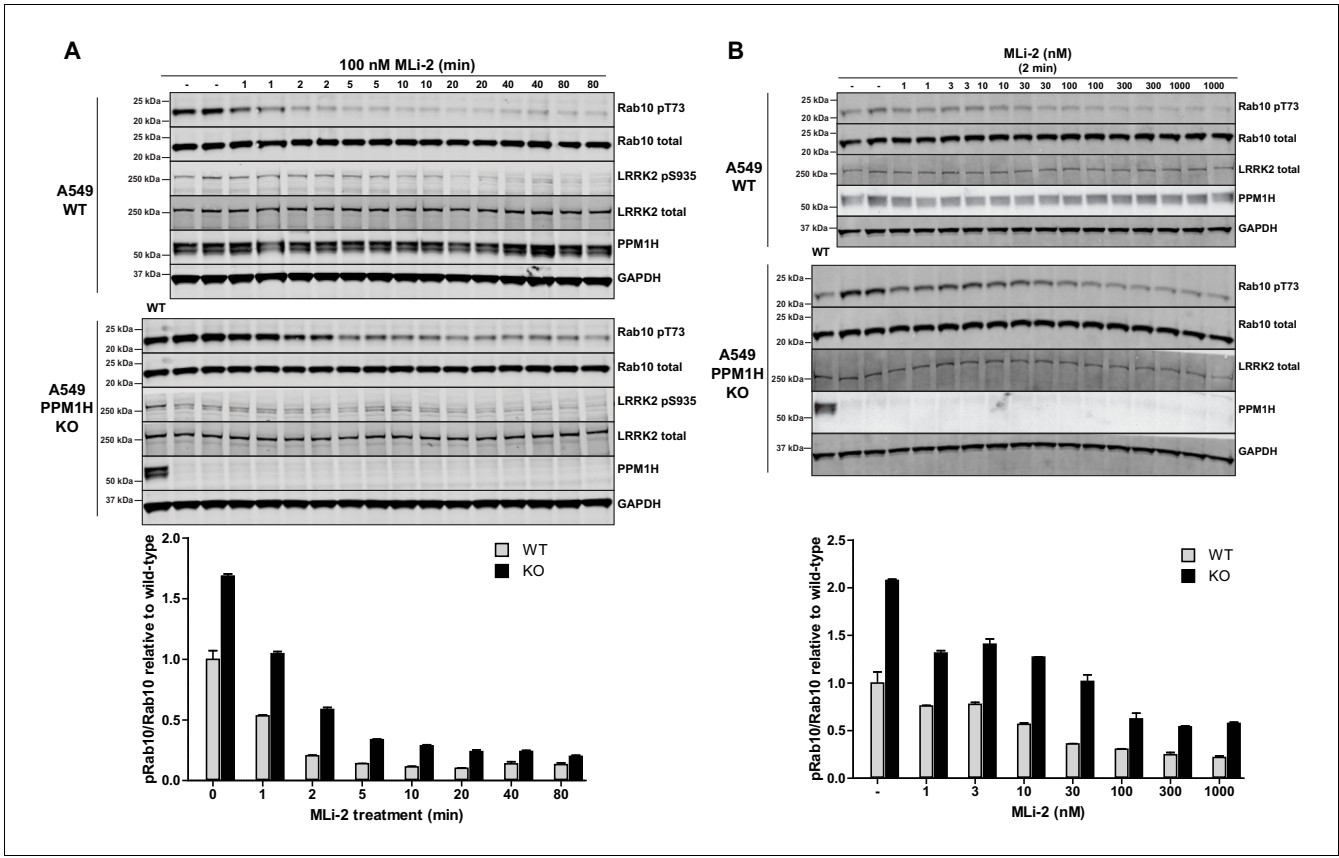

**Figure 7.** MLi-2 inhibitor time-course and dose-response in wild-type and PPM1H knock-out A549 cells. (**A**) Wild-type and PPM1H knock-out A549 cells were treated ±100 nM MLi-2 inhibitor for the indicated times prior to cell lysis. 20 μg whole cell lysate was subjected to immunoblot analysis with the indicated antibodies at 1 μg/mL concentration and membranes analyzed using the OdysseyClx Western Blot imaging system. The ratio of phospho-Thr73 Rab10/total Rab10 was quantified using Image Studio software and data presented relative to the phosphorylation ratio observed in wild-type cells. Each lane represents cell extract obtained from a different dish of cells (two replicates per time-point). (**B**) Wild-type and PPM1H knock-out A549 cells were treated with indicated concentrations of MLi-2 inhibitor for 2 min. Rab10 phosphorylation was assessed and quantified as in (**A**).

DOI: https://doi.org/10.7554/eLife.50416.014

dephosphorylation observed after 1 min, ~60% after 2 min, ~75% after 5 min and basal levels observed after 10 min (*Figure 7A*). We also performed a dose dependence treatment of MLi-2 using a 2 min time point. This revealed that in wild-type cells ~10 nM MLi-2 induced ~50% dephosphorylation of Rab10 in 2 min (*Figure 7B*). In PPM1H knock-out cells ~30 nM MLi-2 was required to induce a 50% dephosphorylation of Rab10 in 2 min (*Figure 7B*).

## PPM1H efficiently dephosphorylates Rab8A and Rab10 in vitro

To test whether PPM1H dephosphorylates Rab proteins in vitro, we produced recombinant Rab8A [Q67L] locked in the GTP-binding conformation, stoichiometrically phosphorylated at Thr72 (*Figure 8—figure supplement 1* and *2*), and recombinant His-SUMO-tagged PPM1H expressed in *E. coli*. Phos-tag SDS-PAGE analysis was used to resolve phosphorylated and dephosphorylated Rab8A (*Ito et al., 2016*), providing a means to monitor direct dephosphorylation of Rab8A by recombinant PPM1H in biochemical experiments (*Figure 8*). Incubation of 2.5 µg phosphorylated Rab8A[Q67L] with PPM1H phosphatase for 30 min induced a dose dependent dephosphorylation with complete dephosphorylation of Rab8A[Q67L] observed with >40 ng of PPM1H and ~50% dephosphorylation observed with 8 ng of PPM1H (*Figure 8A*). In parallel experiments, addition of 0.2 µg of catalytically inactive PPM1H[H153D] or catalytically inactive substrate trapping mutant PPM1H[D288A] (discussed below) markedly inhibited dephosphorylation of Rab8A[Q67L] (*Figure 8A*). PPM1H (40 ng) induced a time dependent dephosphorylation of Rab8A[Q67L]: 50% dephosphorylation was achieved within 10 min, and the reaction proceeded to completion by 80 min (*Figure 8B*). We also compared the rate that PPM1H dephosphorylated wild-type Rab8A complexed to GDP and GTPγS and found that both forms were similarly dephosphorylated (*Figure 8A and B*). In contrast, 0.2 µg PPM1M or PPM1J did not significantly dephosphorylate Rab8A[Q67L] under the same conditions in which 40 ng PPM1H quantitatively dephosphorylated Rab8A[Q67L] (*Figure 8C and D*).

Analogous experiments were performed using wild-type Rab10 locked in the GDP conformation that we were able to purify phosphorylated at Thr73 at a stoichiometry of about 60% (*Figure 8—figure supplements 3,4*). Dose dependent activity of PPM1H towards phosphorylated Rab10 followed a similar pattern to Rab8A, with complete dephosphorylation of 2.5 µg phospho-Rab10 observed after 30 min of incubation with >40 ng of PPM1H (*Figure 8—figure supplement 5A*). Time dependent activity assay of PPM1H towards Rab10 showed complete dephosphorylation after 80 min of incubation of 2.5 µg phospho-Rab10 with 40 ng of PPM1H (*Figure 8—figure supplement 5B*). 0.2 µg of purified PPM1J and PPM1M were also observed to weakly dephosphorylate Rab10 *Figure 8—figure supplement 5C D*).

## Identification of a substrate trapping PPM1H mutant

A recent study reported the crystal structure of a "substrate trapped" catalytically inactive mutant of PPM1A complexed with a phosphorylated peptide substrate (*Debnath et al., 2018*). This substrate trapping conformation was achieved by mutation of an Asp residue (Asp146 in PPM1A) that is conserved in 16 out of 20 PPM family phosphatases. We therefore explored whether mutation of the equivalent residue in PPM1H (Asp288) would also yield high affinity interaction with phosphorylated Rab proteins. We overexpressed wild-type and PPM1H[D288A], as well as the corresponding mutants of the closely related PPM1J[D279A] and PPM1M[D235A] in HEK293 cells expressing LRRK2 [R1441G]. Phosphatases were immunoprecipitated and immunoblotting revealed that Rab8A and Rab10 co-immunoprecipitated with the PPM1H[D288A] mutant, but not with wild-type PPM1H or the substrate trapping mutants of PPM1J[D279A] or PPM1M[D235A] (*Figure 9A*). Phos-tag immunoblot analysis confirmed that it was the LRRK2 phosphorylated species of Rab8A and Rab10 that co-immunoprecipitated with PPM1H[D288A] (*Figure 9—figure supplement 1A*).

We next analyzed total immunoprecipitates of wild-type PPM1H and PPM1H[D288A] using quantitative TMT-labeling mass-spectrometry. In addition to Rab8A and Rab10, we observed Rab8B is significantly enriched in the PPM1H[D288A] immunoprecipitates. Only two other proteins were significantly enriched in the PPM1H[D288A] immunoprecipitates, namely the metabolic enzymes acetyl-CoA carboxylases 1/2 (ACACA/B) (*Figure 9B*). In subsequent immunoblotting experiments we were unable to confirm specific interaction of ACACA/B with PPM1H[D288A]. Further work is required to assess whether Acetyl-CoA carboxylases physiologically interact with PPM1H.

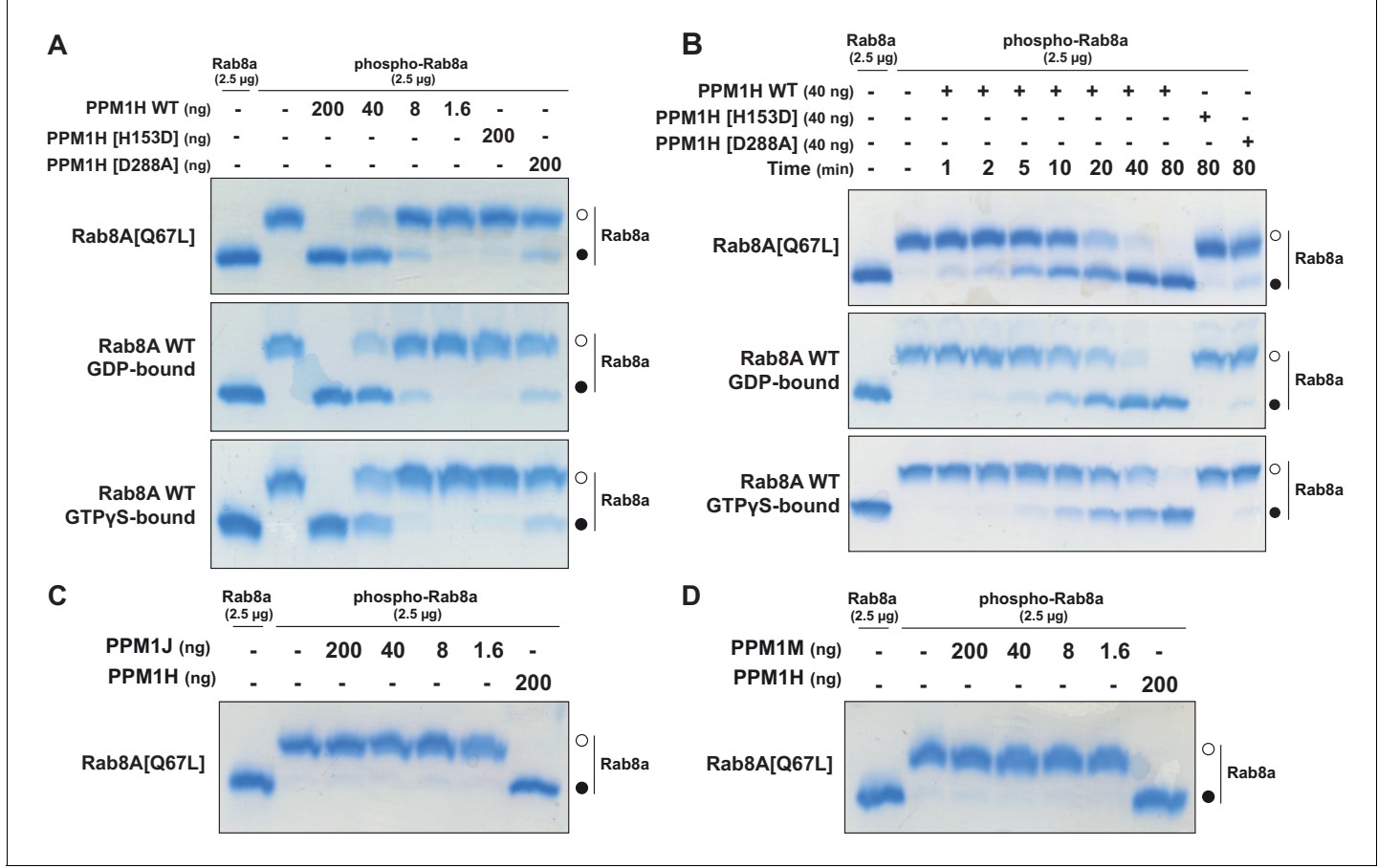

**Figure 8.** PPM1H dephosphorylates Rab8A in vitro. (**A**) The indicated amounts of recombinant wild-type and mutant PPM1H (with a His-Sumo N-terminal tag, expressed in *E. coli*) were incubated in vitro with 2.5 μg pT72 phosphorylated Rab8A[1–181, Q67L-GTP bound conformation], 2.5 μg of Rab8A[1–181 wild-type, GDP-bound] or 2.5 μg of Rab8A[1–181 wild-type, GTP-bound] for 30 min in the presence of 10 mM MgCl₂ in 40 mM HEPES pH 7.0 buffer. Reactions were terminated by addition of SDS Sample Buffer and analyzed by Phos-tag gel electrophoresis that separates phosphorylated and dephosphorylated Rab8A. The gel was stained with Instant Blue Coomassie. Bands corresponding to phosphorylated and non-phosphorylated Rab10 are marked with open (○) and closed (●) circles, respectively. (**B**) As in (**A**) except that a time-course assay was performed using 2.5 μg pT72 phosphorylated Rab8A[1–181, Q67L-GTP bound conformation], 2.5 μg of Rab8A[1–181 wild-type, GDP-bound] or 2.5 μg of Rab8A[1–181 wild-type, GTP-bound] and 40 ng wild-type or mutant PPM1H for the indicated times. (**C**) As in (**A**) except that PPM1J was assessed. (**D**) As in (**A**) except and PPM1M was assessed.

DOI: https://doi.org/10.7554/eLife.50416.015

The following figure supplements are available for figure 8:

**Figure supplement 1.** Purification and characterization of Thr72 phosphorylated Rab8A[1–181, Q67L].
DOI: https://doi.org/10.7554/eLife.50416.016
**Figure supplement 2.** Mass spectrometry characterization of Thr72 phosphorylated Rab8A[1–181, Q67L].
DOI: https://doi.org/10.7554/eLife.50416.017
**Figure supplement 3.** Purification and characterization of Thr73 phosphorylated Rab10[1-181].
DOI: https://doi.org/10.7554/eLife.50416.019
**Figure supplement 4.** Mass spectrometry characterization of Thr73 phosphorylated Rab10[1–181, Wt].
DOI: https://doi.org/10.7554/eLife.50416.020
**Figure supplement 5.** PPM1H dephosphorylates Rab10 in vitro.
DOI: https://doi.org/10.7554/eLife.50416.018
**Figure supplement 6.** Coomassie stained 4-20% gradient SDS polyacrylamide gel electrophoresis analysis of 2.5 μg of purified recombinant phosphatases.
DOI: https://doi.org/10.7554/eLife.50416.021

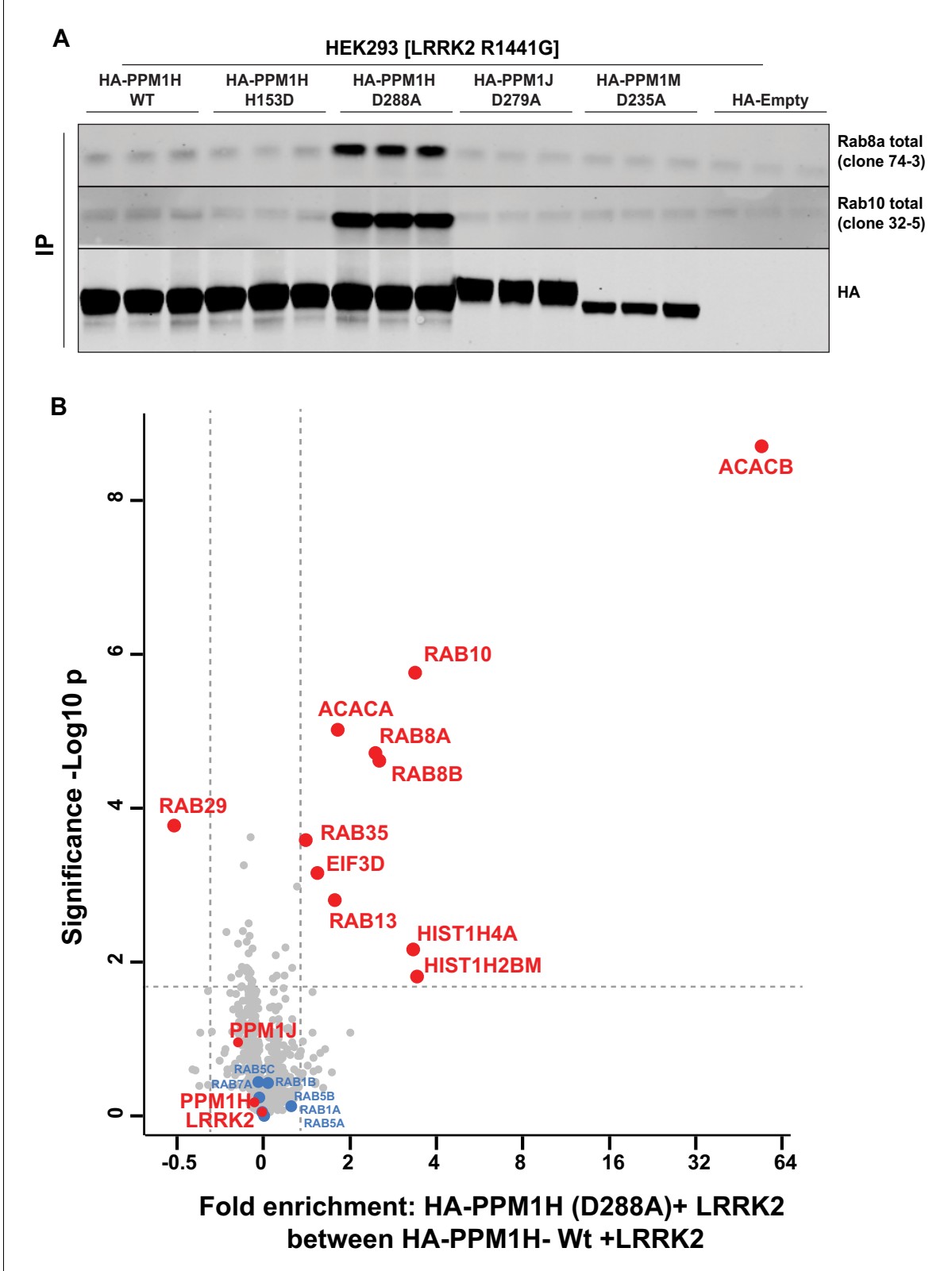

**Figure 9.** Identification of a substrate trapping mutant of PPM1H that forms a stable interaction with endogenous LRRK2 phosphorylated Rab8A and Rab10. (**A**) HEK293 cells were transiently transfected with constructs expressing Flag-LRRK2[R1441G] and either wild-type HA-PPM1H or the PPM1H [D288A] mutant, as well as the corresponding mutants of the closely related PPM1J[D279A] and PPM1M[D235A]. 24 hr post-transfection cells were lysed and subjected to a HA-immunoprecipitation and analyzed by immunoblotting with the indicated antibodies (1 μg/ml). Membranes were developed

*Figure 9 continued on next page*

*Figure 9 continued*
using Odyssey CLx Western Blot imaging. Each lane represents cell extract obtained from a different dish of cells (three replicates per condition). (B) HEK293 cells were transiently transfected with constructs expressing Flag-LRRK2[R1441G] and either wild-type HA-PPM1H or the HA-PPM1H[D288A] mutant, lysed and subjected to a HA-immunoprecipitation and total immunoprecipitates were subjected to 'on-bead' digestion using trypsin following multiplexed TMT labeling the samples were pooled and fractionated into four fractions and analyzed on an Orbitrap Fusion Lumos Tribrid mass spectrometer in MS3 mode. The raw data was processed using MaxQuant pipeline and protein groups were further processed using Perseus software suite. The x-axis of the volcano plot represents the differential enrichment between HA-PPM1H[D288A]+LRRK2[R1441G] and wild-type HA-PPM1H +LRRK2[R1441G] and the y-axis represents the permutation-based false discovery rate corrected significance of a two tailed t-test. The differential enriched protein groups are highlighted in red filled circles along with the protein groups that are not changing are highlighted in blue with their gene names. The list of protein groups is included in *Supplementary file 3*.
DOI: https://doi.org/10.7554/eLife.50416.022
The following figure supplements are available for figure 9:

**Figure supplement 1.** PPM1H[D288A] substrate trapping mutant stably interacts with LRRK2 phosphorylated Rab10 protecting it from dephosphorylation.
DOI: https://doi.org/10.7554/eLife.50416.023
**Figure supplement 2.** PPM1H[D288A] substrate trapping mutant blocks dephosphorylation of overexpressed Rab10 HEK293 cells were transiently transfected with LRRK2[R1441G] and constructs expressing empty HA-vector (no phosphatase control).
DOI: https://doi.org/10.7554/eLife.50416.024

We also co-expressed PPM1H[D288A] with wild-type, pathogenic or catalytically inactive LRRK2. This revealed that significantly higher levels of Rab10 co-immunoprecipitated with PPM1H[D288A] in cells expressing wild-type or pathogenic LRRK2 compared with catalytically inactive LRRK2 (*Figure 9—figure supplement 1B*), consistent with PPM1H[D288A] interacting preferentially with LRRK2-phosphorylated Rab proteins. If PPM1H[D288A] forms a stable complex with phosphorylated Rab proteins in vivo, this would be expected to prevent their dephosphorylation following inhibition of LRRK2. Consistent with this, overexpression of PPM1H[D288A] and to a lesser extent PPM1H [D288E] suppressed dephosphorylation of Rab10 following treatment with 200 nM MLi-2 for 90 min, conditions that induce complete dephosphorylation of Rab proteins in the absence of PPM1H [D288A/E] overexpression (*Figure 9—figure supplement 1C*). Indeed, in cells expressing no phosphatase or the catalytically inactive PPM1H[H153D] mutant, Rab10 was almost completely dephosphorylated following 5-10 min of 200 nM MLi-2 treatment (*Figure 9—figure supplement 2A and B*). However, in cells expressing the PPM1H[D288A] substrate trapping mutant, dephosphorylation was markedly delayed and Rab10 was still significantly phosphorylated after 320 min 200 nM MLi-2 treatment (*Figure 9—figure supplement 2C*).

## PPM1H localizes to the Golgi

We used immunofluorescence microscopy of HA-PPM1H in RPE (retinal pigment epithelium) cells in relation to well characterized marker proteins including GCC185 (trans Golgi network), p115 and beta-COP (cis and medial Golgi), cation independent mannose 6-phosphate receptor (perinuclear late endosomes), and ACBD3 (medial Golgi; *Tie et al., 2016*; *Figure 10*). PPM1H was detected in the cytoplasm and also showed strong co-localization with the medial Golgi protein, ACBD3, as well as cis and medial Golgi markers, as quantified by Pearson's correlation coefficient (*Figure 10*, left); this co-localization was retained upon fragmentation of the Golgi by nocodazole-induced microtubule depolymerization, the gold standard for membrane compartment localization (*Figure 10*, right). In addition, a significant pool of Rab10 co-localized with Golgi-associated PPM1H in RPE cells (*Figure 10—figure supplement 1*) and MEF cells (*Figure 4*). Thus, PPM1H is present in cytosol, and also associates with the cis and medial Golgi complex where it can act on Rab8A and Rab10 proteins.

## PPM1H knockdown suppresses primary cilia formation to the same extent as LRRK2 pathogenic mutations

We have shown previously that pathogenic LRRK2 suppresses cilia formation in cell culture and mouse brain (*Steger et al., 2017*; *Dhekne et al., 2018*) in a process that requires Rab10 and RILPL1 proteins (*Dhekne et al., 2018*). If Rab GTPase phosphorylation contributes to the regulation of cilia formation in wild-type cells, depletion of PPM1H should also increase Rab phosphorylation, thereby

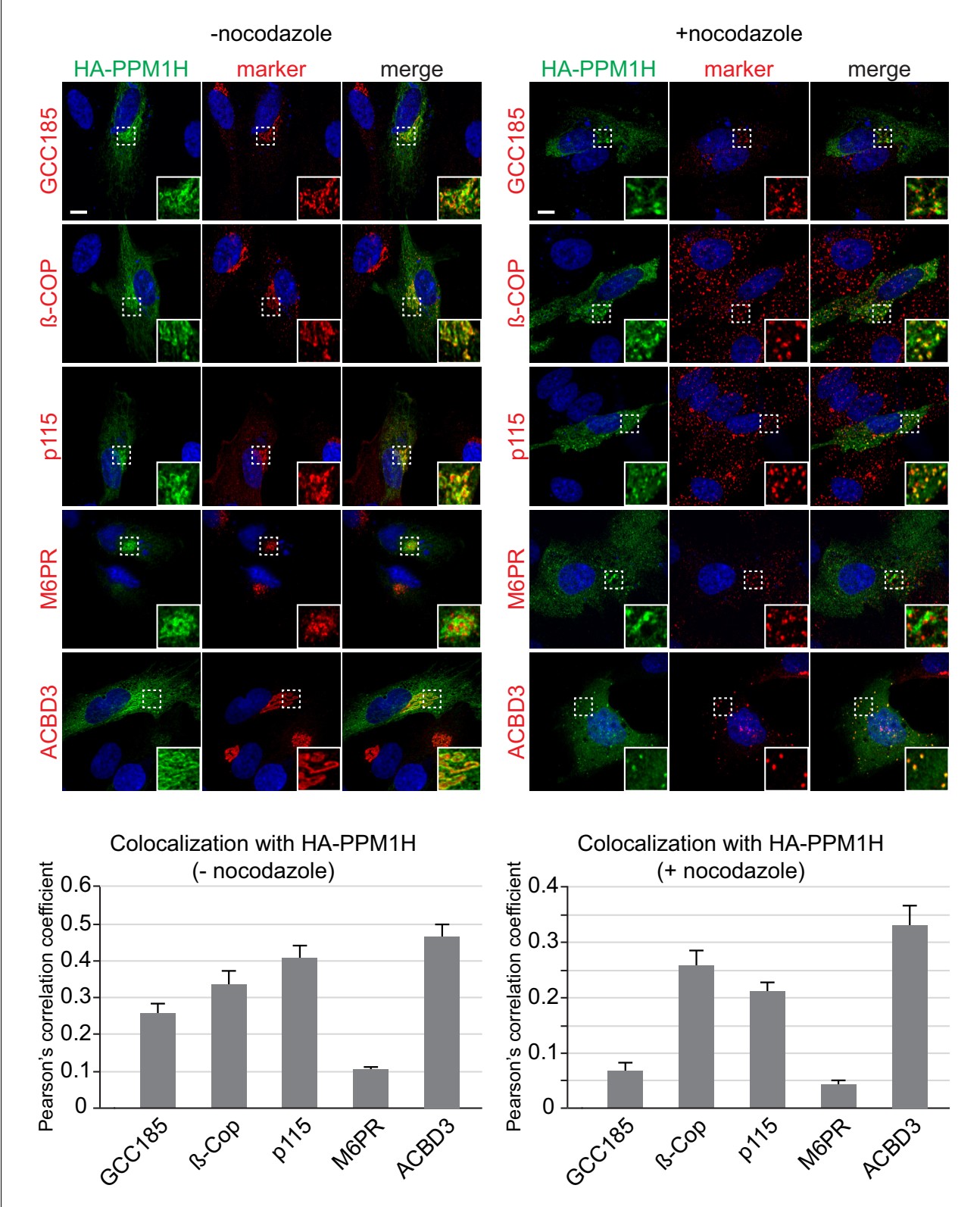

**Figure 10.** Localization of PPM1H in RPE cells. RPE cells were transiently transfected with HA-PPM1H plasmid. After 24 hr, cells were treated ±20 μM nocodazole for 2 hr before fixation. Cells were subsequently stained with mouse or rabbit anti-HA antibody (green) and the following Golgi markers (shown in red): rabbit anti-GCC185, rabbit anti-ß-COP, mouse anti-p115, mouse anti-cation independent mannose 6-phosphate receptor, and rabbit

*Figure 10 continued on next page*

*Figure 10 continued*

anti-ACBD3. Nuclei were stained with DAPI (blue). Scale bar, 10 μm. Shown are maximum intensity projections. For each condition at least 20 cells were analyzed.

DOI: https://doi.org/10.7554/eLife.50416.025

The following figure supplement is available for figure 10:

**Figure supplement 1.** Localization of PPM1H and GFP-Rab10 in RPE cells.

DOI: https://doi.org/10.7554/eLife.50416.026

inhibiting cilia formation. To test this, MEF cells were infected with lentiviruses encoding 3 different shRNAs to deplete PPM1H protein. As shown in *Figure 11*, depletion of PPM1H decreased cilia formation compared with control (scrambled shRNA sequence) infected cells, by a process that required LRRK2, as it was not seen when cells were cultured in the presence of the MLi-2 LRRK2 inhibitor (*Figure 11A*). In MEF cells, endogenous PPM1H levels are below the limit of detection by currently available antibodies, thus depletion in shRNA expressing cells was monitored by qPCR (*Figure 11B*) and were consistent with the phenotypes observed. These experiments show that endogenous PPM1H protein contributes to the regulation of cilia formation in MEF cells and confirms a role for wild-type LRRK2 protein in this important cellular process.

## Discussion

PPM phosphatases (previously known as PP2C phosphatases) are an ancient family of enzymes found in bacteria (*Obuchowski et al., 2000*) and fungi (*Feng et al., 2010*). Plants possess over 80 different PPM phosphatases that play important roles in hormone abscisic acid signaling and protection against stress (*Schweighofer et al., 2004*; *Ma et al., 2009*). PPM phosphatases are evolutionarily unrelated to other serine/threonine protein phosphatases and there are 20 family members in humans (*Figure 5B*) (*Chen et al., 2017*). These enzymes are characterized by their requirement for supplementation with millimolar concentrations of $Mg^{2+}$ or $Mn^{2+}$ ions for activity in vitro and are also insensitive to inhibition by okadaic acid or calyculin-A (*Shi, 2009*). The structure of human PPM1A displays two tightly bound $Mn^{2+}$ ions in the active site and a small subdomain, termed the Flap, located adjacent to the active site (*Das et al., 1996*). Crystal structures of bacterial (*Bellinzoni et al., 2007*) and plant (*Melcher et al., 2009*) PPM homologs are nearly identical to that of the human PPM1A phosphatase core domain. In addition, a third, loosely associated magnesium ion, termed M3, appears to be required for optimal catalysis of all PPM family members (*Tanoue et al., 2013*).

PPM1H is a member of a group of 3 PPM phosphatases that have been termed the 'PPM1H subfamily' (*Chen et al., 2017*). This consists of PPM1H (URCC2, ARHCL1, NERPP-2C), PPM1J (PP2Cζ) and PPM1M (PP2Cη). PPM1H is most widely expressed in brain, but also found at varying levels in most other tissues including lung, kidney and spleen (*Figure 12—figure supplement 1A* to 1B). In blood cells, PPM1H is expressed at high levels in basophils and lower levels in neutrophils and is very low or not detected in most other cells including monocytes (*Figure 12—figure supplement 2*). PPM1M, which is also able to dephosphorylate LRRK2-phosphorylated Rab10, was undetectable in most tissues, and only observed at very low levels in the prostate gland and urinary bladder, despite mRNA levels being detected at similar levels to PPM1H in most tissues (*Figure 12—figure supplement 1C* to 1D). In the blood, PPM1M is most highly expressed in neutrophils (*Figure 12—figure supplement 1C* to 1D). Consistent with high expression in the brain, PPM1H was first discovered in a screen to identify protein phosphatases that are highly expressed in neuronal cell lines and in rat brains (*Labes et al., 1998*). In future studies, it will be important to explore whether high levels of PPM1H, or low levels of LRRK2 expression in neuronal cells account for the relatively low levels of LRRK2 phosphorylated Rab10 in brain in comparison with other tissues (*Lis et al., 2018*; *Fan et al., 2018*).

The PPM1H subfamily phosphatases are conserved in animals from sponge to human (*Chen et al., 2017*); invertebrates from sponge to ciona encode a single family member. The three copies found in mammals probably arose by two independent duplication events (*Chen et al., 2017*). Human PPM1H possesses 514 residues encompassing a PPM type phosphatase domain with no other known functional motifs (*Chen et al., 2017*). In previous work PPM1H was also reported to

be overexpressed in colon adenocarcinoma and co-immunoprecipitated with the CSE1L apoptosis regulator that was suggested to comprise a substrate for PPM1H (*Sugiura et al., 2008*). Another study found PPM1H confers trastuzumab resistance by promoting the dephosphorylation of p27 CDK inhibitor at threonine 187 (*Lee-Hoeflich et al., 2011*). It has also been suggested that patients whose tumors express low levels of PPM1H have reduced survival from colorectal cancer (*Xu et al., 2019*).

Our data reveal that PPM1H specifically dephosphorylates a subset of Rab proteins phosphorylated at the effector-binding motif by LRRK2, namely Rab8A, Rab8B, Rab10 and Rab35. PPM1H fails to dephosphorylate Rab8A protein phosphorylated at the distinct Ser111 site regulated by the PINK1 kinase, also implicated in PD (*Figure 3—figure supplement 2A*) (*Lai et al., 2015*). The specificity of PPM1H for phosphorylated Rab proteins is also highlighted by the substrate trapping PPM1H[D288A] mutant that binds to endogenous LRRK2 phosphorylated Rab8A, Rab8B and Rab10 with sufficient affinity to permit co-immunoprecipitation and suppress dephosphorylation (*Figure 9*). We also observed that PPM1H efficiently dephosphorylated Rab8A complexed to either GDP or GTPγS (*Figure 8A and B*), suggesting that it can recognize both these conformations. A priority for future work will be to elucidate the mechanism by which specificity of PPM1H for dephosphorylating LRRK2 phosphorylated Rab proteins is achieved. Structural analysis of the substrate-trapping mutant of PPM1H complexed to phosphorylated Rab8A will be highly informative. It will also be interesting to determine whether PPM1H possesses regulatory subunits that could control its activity, specificity and localization.

PPM1H is located both in cytosol and on the cis and medial Golgi (*Figure 10*) and further work is needed to elucidate how PPM1H localizes to the Golgi and how it might translocate to other membranes and vesicles where LRRK2 phosphorylated Rab proteins reside. It is interesting to note that there is a significant pool of Rab10 on the Golgi, and Rab8A on the Golgi and perinuclear compartments (*Figure 4*; *Figure 10—figure supplement 1*; *Dhekne et al., 2018*). Perhaps PPM1H's Golgi localization serves to maintain those pools of Rab8A and Rab10 in functional forms to be able to perform functions other than cilia regulation. Another important goal will be to analyze the relationship between PPM1H localization and LRRK2 substrate selection in cells.

The PPM1H-related enzyme, PPM1M (*Figure 5B*) also dephosphorylated Rab10 and Rab8A when overexpressed in cells, albeit to a lower extent than PPM1H (*Figure 5A*). In corresponding biochemical studies, we also found that PPM1M displayed weak activity towards phosphorylated Rab10 (*Figure 8—figure supplement 5D*), but did not dephosphorylate Rab8A significantly (*Figure 8D*). In overexpression experiments PPM1J phosphatase displayed no activity towards dephosphorylating Rab10 (*Figure 5A*) although weak activity was observed in in vitro biochemical analyzes (*Figure 8—figure supplement 5C*). Previous work revealed that at least 22 Rab proteins not regulated by LRRK2 are phosphorylated at their effector motifs (*Steger et al., 2017*), including Rab7A phosphorylated at Ser72 by TBK1 (*Heo et al., 2018*) or LRRK1 (*Hanafusa et al., 2019*), and Rab1 that is phosphorylated at Thr75 by TAK1 (*Levin et al., 2016*). It will be interesting to probe whether PPM1M, PPM1J or indeed PPM1H might dephosphorylate other Rab proteins. Out of the other 11 PPM family phosphatases tested, only PHLPP1 (PH domain leucine-rich repeat protein phosphatase) displayed weak ability to dephosphorylate Rab10 in an overexpression assay (*Figure 5C*).

As LRRK2 inhibitors still induce dephosphorylation of Rab10 in PPM1H knock-out A549 cells (*Figure 7A and B*), other protein phosphatase(s), that could include PPM1M, must also control Rab10 dephosphorylation in vivo. Our analysis also revealed that knock-down of a number of other phosphatases (TAB1, PTPN6, PTPN22, PTPN23, MTMR9, NT5DC2, DUSP3, DUSP18, PLPP6) modestly increased Rab10 protein phosphorylation (*Figure 2D and E*). These phosphatases include tyrosine phosphatases (PTPN6, PTPN22, PTPN23), dual specificity phosphatases (DUSP3, DUSP18), a lipid phosphatase (MTMR9), a catalytically inactive pseudophosphatase (TAB1), and metabolic phosphatases (NT5DC2 and PLPP6). None of these enzymes are known to dephosphorylate Ser/Thr residue phosphorylated proteins, and it is possible that the observed actions of these phosphatases are indirect; further analysis will be required to probe any physiological significance. Interestingly, we also found that knock-down of a small group of phosphatases (GAK, ALPI, PPEF2, MDP1, PUDP) reduced LRRK2 levels leading to an inhibition of Rab10 protein phosphorylation (*Figure 2F*). Mutations in GAK (also known as Auxilin-2) have been linked to PD, as revealed by genome-wide association studies (*Nalls et al., 2014*). Thus, it will be interesting to probe the mechanisms by which GAK, as well as the other phosphatases, impinge on the LRRK2 pathway.

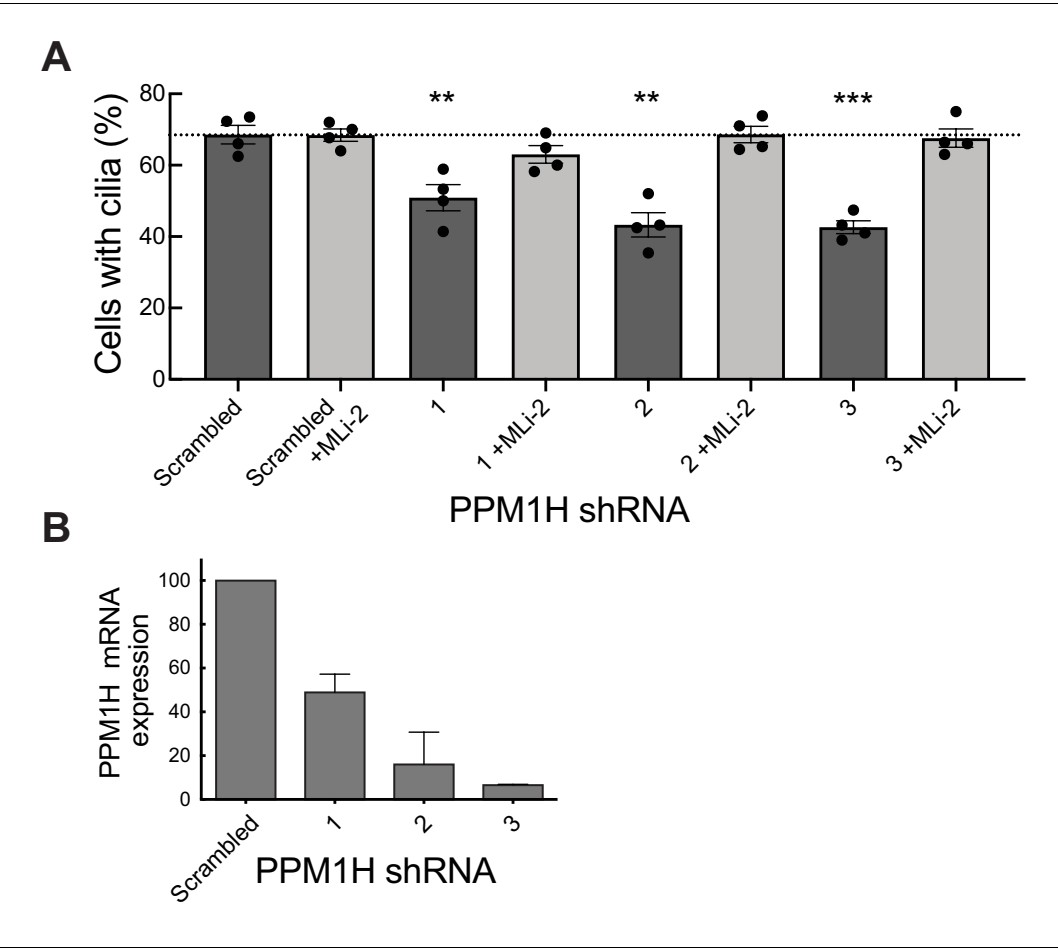

**Figure 11.** PPM1H depletion in wild-type MEF cells decreases ciliation, consistent with increased Rab phosphorylation. Cells expressing the indicated shRNAs were assayed for ciliation 24 hr after serum starvation. (**A**) Shown is the average of duplicate determinations made at identical cell confluency; error bars represent SEM. The experiment was replicated four times by two lab members; p=0.008 (**), 0.001(**) or 0.0002 (***) for shRNAs 1, 2 and 3, respectively (unpaired two-tailed t test, GraphPad PRISM software). (**B**) PPM1H mRNA levels detected in the corresponding cell extracts using two different primers by qPCR in triplicate in relation to a GAPDH control. Error bars represent SD.

DOI: https://doi.org/10.7554/eLife.50416.027

In summary, our data provide compelling evidence that PPM1H acts to dephosphorylate Rab proteins in vivo and therefore counteracts LRRK2 signaling (*Figure 12*). This is supported by the findings that knock-down (*Figure 3A*) as well as knock-out (*Figure 6*) of PPM1H induce endogenous Rab10, Rab8A and Rab35 protein phosphorylation and PPM1H directly dephosphorylates phosphorylated Rab8A and Rab10 in vitro (*Figure 8*). Overexpression of PPM1H in cells abolishes phospho-Rab10 staining of wild-type MEF cells (*Figure 4*). We identify a substrate trapping mutant of PPM1H that binds to endogenous LRRK2 phosphorylated-Rab8A as well as Rab10 and also protects Rab10 from becoming dephosphorylated following inhibition of LRRK2 (*Figure 9*). LRRK2 has been reported to regulate many biological responses such as autophagy (*Plowey et al., 2008*), immune responses (*Moehle et al., 2012*) as well as phagosome maturation (*Härtlova et al., 2018*), and a key question is whether LRRK2 controls these effects by phosphorylating Rab proteins or via an independent pathway. This issue of whether Rab proteins are involved in mediating the actions of LRRK2 can be addressed in the future by probing the impact of manipulating PPM1H levels. Indeed, we have used this approach to demonstrate that knock-down of PPM1H blocks primary cilia formation in a manner analogous to pathogenic, activating LRRK2 mutations (*Figure 11*). This is consistent with our previous work demonstrating that the ability of LRRK2 to inhibit primary cilia formation is indeed

dependent on Rab8A and Rab10 protein phosphorylation (*Dhekne et al., 2018*). In future work it will be important to explore whether PPM1H contributes to PD risk. For example, pathogenic mutants of LRRK2 are incompletely penetrant and a majority of patients may not develop PD (*Ruiz-Martínez et al., 2010*; *Goldwurm et al., 2011*). It would be critical to explore whether increased expression or activity of PPM enzymes protects patients with LRRK2 mutations from developing PD by enhancing Rab protein dephosphorylation. Conversely, reduced expression or activity of PPM1H phosphatase would be expected to promote Rab protein phosphorylation and enhance PD risk. Targeting PPM1H to increase its activity or expression in order to promote Rab protein dephosphorylation could be explored as a therapeutic strategy for preventing and/or treating LRRK2-mediated PD.

# Materials and methods

## Key resources table

| Reagent type (species) or resource | Designation | Source or reference | Identifiers | Additional information |
|---|---|---|---|---|
| Cell line (*Homo sapiens*) | Lung adenocarcinoma | ATCC | RRID:CVCL_0023 | |
| Cell line (*M. musculus*) | LRRK2[R1441C], Mouse embryonic fibroblasts | *Steger et al., 2017* | | |
| Antibody | anti-phospho-Rab10 (Thr73) (rabbit monoclonal) | Abcam | #ab230261 | WB (1:1000) |
| Antibody | anti-phospho-Rab8 (Thr72) (rabbit monoclonal) | Abcam | #ab230260 | WB (1:1000) |
| Antibody | anti-phospho-Rab12 (Ser106) (rabbit monoclonal) | Abcam | *Mir et al., 2018* | WB (1:1000) |
| Antibody | anti-Rab8A (rabbit monoclonal) | Abcam | #ab237702 | WB (1:1000) |
| Antibody | anti-Rab10 (mouse monoclonal) | nanoTools | 0680–100/Rab10-605B11 | WB (1:1000) |
| Antibody | anti-PPM1H (sheep polyclonal) | MRC PPU Reagents and Services | DA018 | WB (1 µg/ml) |
| Antibody | anti-Akt (sheep polyclonal) | MRC PPU Reagents and Services | S695B | WB (1 µg/ml) |
| Antibody | anti-Rab12 (sheep polyclonal) | MRC PPU Reagents and Services | SA227 | WB (1 µg/ml) |
| Antibody | anti-Rab43 (sheep polyclonal) | MRC PPU Reagents and Services | S417B | WB (1 µg/ml) |
| Antibody | anti-S6K1 (sheep polyclonal) | MRC PPU Reagents and Services | SA135 | WB (1 µg/ml) |
| Antibody | anti-phospho-Ser935 LRRK2 (rabbit monoclonal) | MRC PPU Reagents and Services | UDD2 | WB (1:1000) |
| Antibody | anti-LRRK2 (rabbit monoclonal) | MRC PPU Reagents and Services | UDD3 | WB (1:1000) |
| Antibody | anti-LRRK2 (mouse monoclonal) | Neuromab | 75–253 | WB (1:1000) |
| Antibody | anti-Phospho-AMPKα (Thr172) (rabbit monoclonal) | CST | #2535, RRID:AB_331250 | WB (1:1000) |
| Antibody | anti-AMPKα (rabbit monoclonal) | CST | #5831, RRID:AB_10622186 | WB (1:1000) |

*Continued on next page*

*Continued*

| Reagent type (species) or resource | Designation | Source or reference | Identifiers | Additional information |
|---|---|---|---|---|
| Antibody | anti-Phospho-Akt Ser473 (rabbit monoclonal) | CST | #9271, RRID:AB_329825 | WB (1:1000) |
| Antibody | anti- Rab35 (rabbit monoclonal) | CST | #9690, RRID:AB_11178805 | WB (1:1000) |
| Antibody | anti- Rab3A (rabbit monoclonal) | CST | #12214 | WB (1:1000) |
| Antibody | anti- ACC1 (rabbit monoclonal) | CST | #3676, RRID:AB_2219397 | WB (1:1000) |
| Antibody | anti- ACC2 (rabbit monoclonal) | CST | #8578, RRID:AB_1094989 | WB (1:1000) |
| Antibody | anti-phospho-S6K1 Thr389 (rabbit monoclonal) | CST | #9205, RRID:AB_330944 | WB (1:1000) |
| Antibody | anti-phospho-MYPT1 Thr853 (rabbit monoclonal) | CST | #4563, RRID:AB_1031185 | WB (1:1000) |
| Chemical compound, drug | MLi-2 | University of Dundee | *Fell et al., 2015* | |
| Chemical compound, drug | Calyculin A | Abcam | #ab141784 | |
| Chemical compound, drug | Okadaic Acid | Abcam | #ab120375 | |
| Chemical compound, drug | Microcystin-LR | Enzo Life Sciences | ALX-350–012 | |
| Transfected construct (human) | HA-EMPTY, pCMV5D | MRC PPU Reagents and Services | DU44059 | |
| Transfected construct (human) | HA-Rab10, pCMV5D | MRC PPU Reagents and Services | DU44250 | |
| Transfected construct (human) | HA-Rab8A, pCMV5D | MRC PPU Reagents and Services | DU35414 | |
| Transfected construct (human) | HA- Rab8B, pCMV5D | MRC PPU Reagents and Services | DU39856 | |
| Transfected construct (human) | HA- Rab12, pCMV5D | MRC PPU Reagents and Services | DU48963 | |
| Transfected construct (human) | HA- Rab3A, pCMV5D | MRC PPU Reagents and Services | DU51539 | |
| Transfected construct (human) | HA- Rab3B, pCMV5D | MRC PPU Reagents and Services | DU55007 | |
| Transfected construct (human) | HA- Rab3C, pCMV5D | MRC PPU Reagents and Services | DU55048 | |
| Transfected construct (human) | HA- Rab3D, pCMV5D | MRC PPU Reagents and Services | DU26388 | |
| Tansfected construct (human) | HA- Rab35, pCMV5D | MRC PPU Reagents and Services | DU26478 | |
| Transfected construct (human) | HA- Rab43, pCMV5D | MRC PPU Reagents and Services | DU26392 | |
| Transfected construct (human) | HA- PPM1H, pCMV5D | MRC PPU Reagents and Services | DU62789 | |
| Transfected construct (human) | HA- PPM1H[H153D], pCMV5D | MRC PPU Reagents and Services | DU62928 | |
| Transfected construct (human) | HA- PPM1H [D288A], pCMV5D | MRC PPU Reagents and Services | DU62985 | |
| Transfected construct (human) | HA- PPM1H [D288E], pCMV5D | MRC PPU Reagents and Services | DU62986 | |

*Continued on next page*

*Continued*

| Reagent type (species) or resource | Designation | Source or reference | Identifiers | Additional information |
|---|---|---|---|---|
| Transfected construct (human) | FLAG-LRRK2 [R1441G], pCMV5D | MRC PPU Reagents and Services | DU13077 | |
| Peptide, recombinant protein | His-SUMO-PPM1H | MRC PPU Reagents and Services | Expressed from DU62835 | |
| Peptide, recombinant protein | His-SUMO-PPM1H[H153D] | MRC PPU Reagents and Services | Expressed from DU68104 | |
| Peptide, recombinant protein | His-SUMO-PPM1H[D288A] | MRC PPU Reagents and Services | Expressed from DU68087 | |

All Plasmids, antibodies and proteins (including datasheets and sequence information) that we have generated for this study can be requested from MRC PPU Reagents and Services (https://mrcppure-agents.dundee.ac.uk/).

## Cell culture and lysis

Wild-type and homozygous LRRK2 R1441C knock-in MEF cells isolated from littermate-matched mouse embryos at day E12.5 (described in *Steger et al., 2017*) were grown in DMEM containing 10% (by vol) fetal calf serum, 2 mM L-glutamine, 100 U/ml penicillin, and 100 µg/ml streptomycin supplemented with non-essential amino acids and 1 mM sodium pyruvate. A549, RPE and HEK293 cells were purchased from ATCC and cultured in DMEM containing 10% (by vol) fetal calf serum, 2 mM l-glutamine, 100 U/ml penicillin, and 100 µg/ml streptomycin. All cells were grown at 37°C, 5% (by vol) $CO_2$ in a humidified atmosphere and regularly tested for mycoplasma contamination. Unless otherwise indicated cells were lysed in an ice-cold lysis buffer containing 50 mM Tris–HCl, pH 7.4, 1% (by vol) Triton X-100, 10% (by vol) glycerol, 0.15 M NaCl, 1 mM sodium orthovanadate, 50 mM NaF, 10 mM 2-glycerophosphate, 5 mM sodium pyrophosphate, 1 µg/ml microcystin-LR, and complete EDTA-free protease inhibitor cocktail (Roche). Lysates were clarified by centrifugation at 20 800 g at 4°C for 10 min and supernatants were quantified by Bradford assay.

## Antibodies

To generate a polyclonal PPM1H antibody a sheep was immunized with full length PPM1H protein (His-SUMO tag cleaved off) followed by 3 further injections 28 days apart, with bleeds performed seven days after each injection. Antibodies were affinity-purified from serum using PPM1H protein. For immunoblotting analysis, the PPM1H sheep antibody was used at a final concentration of 1 µg/ml. We noted that incubation for 90 min at room temperature appeared to provide clearer results than overnight incubation at 4°C. The sheep PPM1H antibody is available from MRC PPU Reagents and Services (sheep number DA018 https://mrcppureagents.dundee.ac.uk/). Sheep polyclonal antibodies detecting total Akt (S695B), total Rab12 (SA227) and total Rab43 (SA135) were purified at the University of Dundee. The Phospho-AMPKα (Thr172) (CST (40H9) #2535), AMPKα total (CST (D5A2) #5831) and Phospho-Akt Ser473 (CST #9271), Rab35 (CST #9690), Rab3A (CST #12214), ACC1 (CST #3676) and ACC2 (CST #8578) were purchased from Cell Signaling Technology. Mouse anti-GAPDH total was from Santa Cruz Biotechnology (sc-32233). PINK1 total antibody was from Novus (BC100-494), OPA1 antibody was from BD (612607). Rabbit monoclonal antibodies for total LRRK2 (N-terminus) (UDD3) and phospho-Ser935 LRRK2 (UDD2) were expressed and purified at University of Dundee as described previously (*Ran et al., 2013a*). The C-terminal total LRRK2 mouse monoclonal antibody was from Neuromab (75-253). The recombinant MJFF-pRab10 (Thr73), MJFF-pRab8 (Thr72) and total Rab8A rabbit monoclonal antibodies were recently described (*Lis et al., 2018*) and are available from Abcam (#ab230261, ab230260, ab237702). The rabbit polyclonal Rab8A phospho-S111 antibody was obtained from Miratul Muqit (University of Dundee). Recombinant antibodies were used at 1 µg/ml final concentration for immunoblotting. The MJFF-total Rab10 mouse antibody was from nanoTools (0680–100/Rab10-605B11, www.nanotools.de) (*Lis et al., 2018*) and was used at 1 µg/ml final concentration. The phospho-Rab12 (S106) rabbit monoclonal antibody was described earlier (*Mir et al., 2018*). Sheep polyclonal antibody detecting total S6K1 was purified at the University of Dundee (S417B). The anti-phospho-S6K1 Thr389 and the anti-

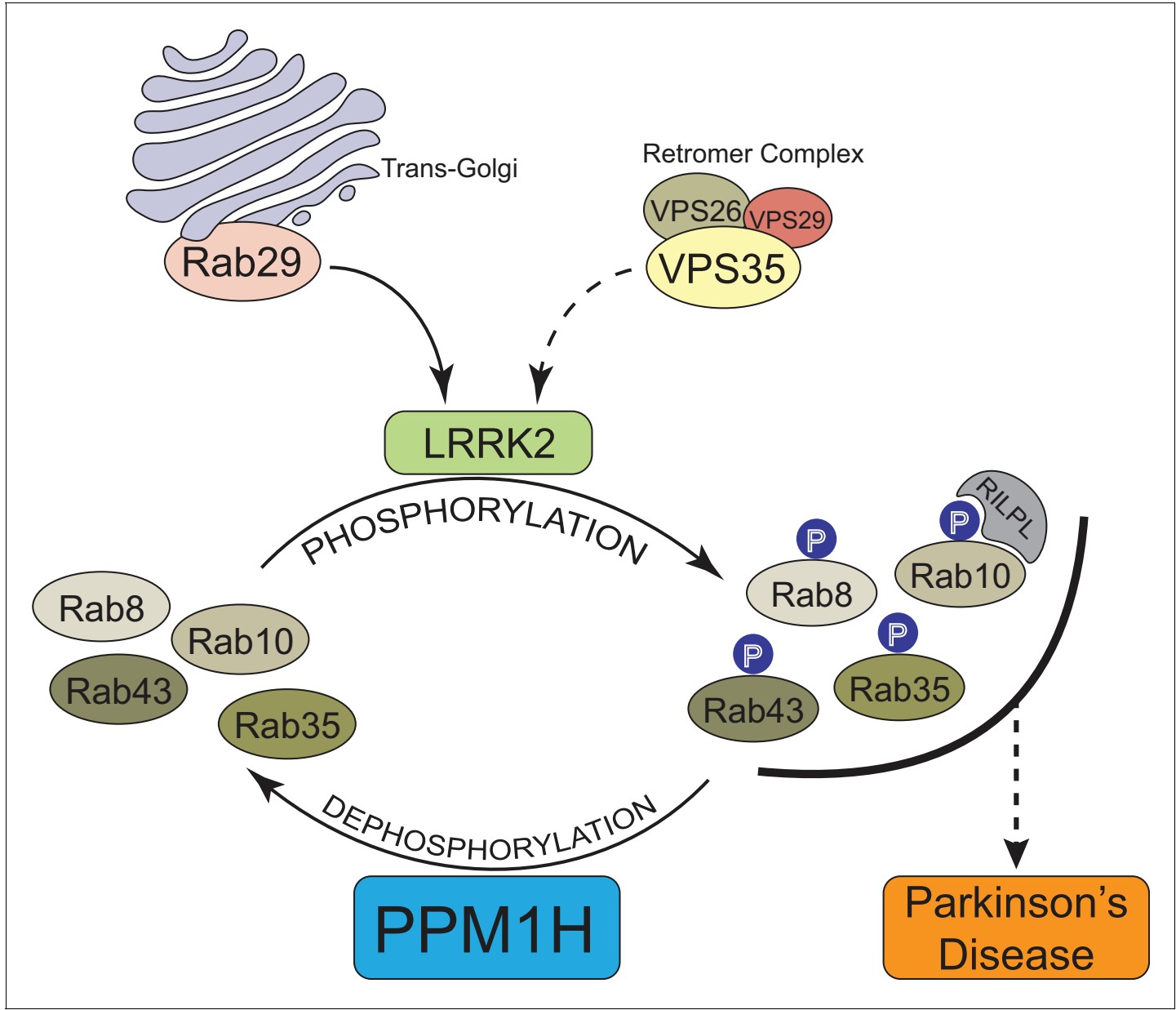

**Figure 12.** Model of how PPM1H counteracts LRRK2 signaling pathway by dephosphorylating Rab proteins.

DOI: https://doi.org/10.7554/eLife.50416.028

The following figure supplements are available for figure 12:

**Figure supplement 1.** The abundance of PPM1H and PPM1M mRNA and protein in human tissues was analyzed using data downloaded from ProtemicsdB public repository database (https://www.proteomicsdb.org/) (*Schmidt et al., 2018*).

DOI: https://doi.org/10.7554/eLife.50416.029

**Figure supplement 2.** The abundance of PPM1H and PPM1M proteins in immune cells isolated from human blood using data available from the immprot database (http://www.immprot.org) (*Rieckmann et al., 2017*).

DOI: https://doi.org/10.7554/eLife.50416.030

phospho-MYPT1 Thr853 antibodies were purchased from Cell Signaling Technology (#9205 and #4563, respectively). For immunoblotting applications all commercial monoclonal antibodies were diluted in 5% (by vol) bovine serum albumin in TBS-T (20 mM Tris base, 150 mM Sodium Chloride (NaCl), 0.2% (by vol) Tween20), and sheep polyclonal antibodies were diluted in 5% (by vol) milk in TBS-T.

## Reagents

MLi-2 inhibitor was synthesized by Natalia Shpiro (University of Dundee) as described previously (*Fell et al., 2015*). Calyculin A (ab141784) and Okadaic Acid (ab120375) were purchased from Abcam. Microcystin-LR was obtained from Enzo Life Sciences (ALX-350-012).

## Plasmids

The following cDNA constructs were used for transfections: HA-Empty (DU44059), HA-Rab10 (DU44250), HA-Rab8A (DU35414), HA-Rab8B (DU39856), HA-Rab12 (DU48963), HA-Rab3A (DU51539), HA-Rab3B (DU55007), HA-Rab3C (DU55048), HA-Rab3D (DU26388), HA-Rab35 (DU26478), HA-Rab43 (DU26392), HA-PPM1H (DU62789), PPM1H[H153D] (DU62928), PPM1H [D288A] (DU62985), HA-PPM1H[D288E] (DU62986), Flag-LRRK2[R1441G] (DU13077). These are available from MRC PPU Reagents and Services (https://mrcppureagents.dundee.ac.uk/). DNA constructs were amplified in *Escherichia coli* DH5α and purified using a Hi-Speed Plasmid Maxi Kit (Qiagen).

## siRNA screens

The siRNA screen was performed using a human siRNA library (Dharmacon) designed to target 322 phosphatases. The list of siRNA targets and the sequences of all siRNA oligonucleotides used are provided in Supplementary File 1. A549 cells were seeded in 6-well plates at 150,000 cells/well. After 24 h cells were transfected using 2 µl Lipofectamine RNAi Max and 20 pmol of siRNA per well. Cells were then cultured for a further 72 h. In Screen 1 and 2, cells were directly lysed without further treatment, whereas in Screen 3, cells were treated for 5 min with 100 nM MLi-2 prior to lysing. Lysates were centrifuged at 20,800 g for 15 min at 4°C, quantified by Bradford assay (Thermo Scientific) and subjected to immunoblot analysis.

## Heavy synthetic peptides

Heavy phosphorylated either 13C615N4 (Arg*) or 13C615N2 (Lys*) containing pRab1(FRpTITS-SYYR*), pRab3 (YRpTITTAYYR*), pRab8(FRpTITTAYYR*), pRab10(FHpTITTSYYR*), total Rab10 (NIDE-HANEDVER*, AFLTLAEDILR*) non-phosphorylated Thr73 pRab10(FHTITTSYYR*), pRab35 (FRpTITSTYYR*) and pRab43(FRpTITQSYYR*) peptides were synthesized from JPT innovative peptide technologies (https://www.jpt.com/). All of the synthesized peptides are of >95% isotopic purity and an independent verification for the absolute amounts were determined by Amino acid analysis (AAA analysis), HPLC and LC-MS/MS analysis.

## Generation of PPM1H CRISPR/Cas9 knockout

CRISPR was performed using a paired nickase approach to minimize off-target effects (*Ran et al., 2013a*). Analysis of the *PPM1H* locus (ENSG00000111110) showed the expression of a single verified transcript (NM_020700.2, ENST00000228705.7) and exon one specific guide pairs with low combined off-targeting scores were subsequently identified using the Sanger Institute CRISPR webtool (http://www.sanger.ac.uk/htgt/wge/find_crisprs). Complementary oligos for the optimal guide pair A (G1 5'- gCAATTTCATGGGCGGCATCA and G2 5'- gCTCGAGTGAGCATATTACTC) were designed and annealed according to the Zhang method (*Ran et al., 2013b*) with BbsI compatible overhangs facilitating cloning into the target vectors; the sense guide G1 was cloned into the puromycin selectable plasmid pBABED P U6 (DU48788, https://mrcppureagents.dundee.ac.uk/) and the antisense guide G2 cloned into the spCas9 D10A expressing vector pX335 (Addgene Plasmid #42335) yielding clones DU64249 and DU64253 respectively.

CRISPR was performed by co-transfecting A549 cells (80% confluency, 10 cm dish) with 3.75 µg of each plasmid using 27 µl of Lipofectamine LTX according to the manufacturer's instructions (Life Technologies). The transfected cells were then incubated for 24 hr in DMEM supplemented with 10% (by vol) FBS, 2 mM L-glutamine, 100 units/ml penicillin and 100 µg/ml streptomycin. Following CRISPR, the medium was replaced with fresh medium supplemented with 3 µg/ml puromycin for a total of 48 hr to enrich for transfectants, adding fresh media each day. The cells were then grown for a further 48 hr, in media without puromycin to provide sufficient cells for single-cell sorting.

Single cells were placed in individual wells of a 96-well plate containing DMEM supplemented with 10% (by vol) FBS, 2 mM L-glutamine, 100 units/ml penicillin, 100 µg/ml streptomycin and 100

µg/ml Normocin (InvivoGen). After reaching ~80% confluency, individual clones were transferred into six-well plates and PPM1H expression determined via immunoblot. Wild-type A549 cells were cultured side-by-side with the knock-out pools and subjected to single cell sorting. four independent wild-type clonal cell lines were obtained and used as controls for knock-out clones.

## Immunoprecipitation assays

Flag-LRRK2[R1441G] and HA-PPM1H, HA-PPM1H[D288A], or HA-empty were transiently overexpressed in HEK293 cells using Polyethylenimine transfection (*Reed et al., 2006*). 24 h post-transfection, cells were lysed in lysis buffer as described above and HA-tagged proteins were immunoprecipitated using anti-HA beads (40 µl resin for 2 mg protein). Immunoprecipitates were then washed twice with lysis buffer containing 0.5 M NaCl and a further three times with Dulbecco's phosphate-buffered saline (200 mg/L Potassium Chloride (KCl), 200 mg/L Potassium Phosphate monobasic (KH2PO4), 8000 mg/L Sodium Chloride (NaCl), 2160 mg/L Sodium Phosphate dibasic (Na2HPO4-7H2O)). 40 µl lithium dodecyl sulfate loading buffer (106 mM Tris HCl, 141 mM Tris Base, 2% (by mass) LDS, 10% (by vol) glycerol, 0.51 mM EDTA, 0.22 mM SERVA Blue G250, 0.175 mM Phenol Red, pH 8.5), diluted 2-fold and then added to the beads. The mixture was then incubated at 100°C for 10 min, and the eluent was collected by centrifugation through a 0.22-µm-pore-size Spin-X column and then supplemented with 2-mercaptoethanol to a final concentration of 1% (by vol). Samples were incubated for 5 min at 70°C before being subjected to immunoblot analysis.

## Sample preparation for TMT experiments

Wild-type and mutant PPM1H was expressed in HEK293 cells, immunoprecipitated using 2 mg of cell lysate, and immunoprecipitates washed as described above. The samples were further subjected to 'on-bead' digestion by adding 150 µl of trypsin digestion buffer (2 M Urea in 50 mM triethylammonium bicarbonate buffer pH 8.0, and 2 mM dithiothreitol, 0.4 µg trypsin) and incubated at 32°C on a Thermomixer with an agitation at 1200 rpm for 30 min. 5 mM iodoacetamide was then added and digestion and agitation continued for a further 16 hr. 1% (by vol) trifluoroacetic acid was added and the resulting mixture centrifuged at room temperature at 2500 g for 2 min. The supernatant was separated from the beads and collected and subjected to Stage-tip based Strong-cation exchange (SCX) as described previously (*Kulak et al., 2014*). Briefly, two SCX disks (Empore SCX 2251. CDS analytical USA) were prepared with 16-gauge needle inserted into a 250 µl pipette tip. SCX tips are activated by adding 100 µl of acetonitrile and centrifuged at 2000 x g for 2 min. Acidified peptide digest was loaded onto the tips and centrifuged at 1500 x g for 5 min and the flow through was reloaded and centrifuged. The stage-tips are washed twice by adding 100 µl of 0.2% (by vol) trifluoroacetic. Peptides were eluted by adding 30 µl of elution buffer (80% (by vol) acetonitrile in 5% (by vol) ammonium hydroxide). Elution step was repeated, and the combined eluates snap frozen and vacuum dried using Speed-Vac.

## TMT labeling

The Vacuum dried peptides were reconstituted in 20 µl of 50 mM triethylammonium bicarbonate pH8.0. 6 µl (80 µg of TMT reporter tag) of each TMT reporter tag (TMT 11 plex kit, Product number: 90110 and A37725., Thermo Fisher Scientific) was added to each sample and incubated at room temperature on a Thermo mixer with a gentle agitation (600 rpm) for 2 hr. The TMT labeling performed corresponds to: HA-PPM1H[D288A]+LRRK2[R1441G] sample 1–4 was labelled to 126, 127N, 127C, 128N and HA-PPM1H-Wt+LRRK2[R1441G] sample 1–4 was labeled to 128C, 129N, 129C, 130N and HA-Empty no PPM1H sample 1–3 was labeled to 130C, 131N and 131C reporter tags. After labeling, another 25 µl of 50 mM triethylammonium bicarbonate pH 8.0 was added to each sample and incubated further 10 min. A 5 µl aliquot from each sample was vacuum dried and desalted using C18 Stage-tip as described above and labeling efficiency established by LC-MS/MS. TMT labeling efficiency was observed to be >99% for each sample. Each sample was quenched by adding 5 µl of 5% (by vol) hydroxyl amine and incubated at room temperature for 10 min. Samples were pooled and vacuum dried in a speed-vac. Pooled TMT labeled peptides were subjected to mini high pH reversed-phase micro-column-based fractionation as described in *Ruprecht et al. (2017)*. A total of 8 fractions were prepared and concatenated into final four fractions (FR1+FR4, FR2+FR5, FR3+FR7 and FR4+FR8), vacuum dried and subjected to LC-MS/MS analysis.

## LC-MS/MS analysis

TMT labeled, and fractionated peptides were reconstituted in 15 µl of 0.1% (by vol) formic acid and 3% (by vol) acetonitrile buffer and analyzed on Orbitrap Fusion Lumos Tribrid mass spectrometer that is interfaced with 3000 RSLC nano liquid chromatography system. Samples were loaded on to a 2 cm trapping column (PepMap C18 100A – 300 µm. Part number: 160454. Thermo Fisher Scientific) at 5 µl/min flow rate using a loading pump and analyzed on a 50 cm analytical column (EASY-Spray column, 50 cm ×75 µm ID, Part number: ES803) at 300 nl/min flow rate that is interfaced to the mass spectrometer using Easy nLC source and electro sprayed directly into the mass spectrometer. We then employed a linear gradient of 3% to 30% of solvent-B at 300 nl/min flow rate (solvent-B: 80% (by vol) acetonitrile) for 105 min and 30% to 40% solvent-B for 15 min and 35% to 99% solvent-B for 5 min which was maintained at 90% B for 10 min and washed the column at 3% solvent-B for another 10 min comprising a total of 140 min run with a 120 min gradient in a data dependent MS3 mode (FT- IT CID MS2-FT MS3). The top 10 precursor ions were targeted for the MS2 and the top 10 fragment ions were isolated by enabling synchronous precursor selection for MS3 and analyzed using an Ultra high-filed Orbitrap mass analyzer. MS1 data was collected and measured using an Ultra high-filed mass analyzer at 120,000 resolution at m/z 200. Precursor ions were isolated using a quadrupole mass filter with 0.7 Da mass window and MS2 spectra were fragmented using collisional induced dissociation and recorded using Ion trap mass analyzer in a rapid mode. Top 10 fragment ions with a 1.2 Da isolation window were isolated and fragmented using beam-type collisional induced dissociation (HCD) at 65% normalized collision energy and analyzed using Ultra-high filed Orbitrap mass analyzer using 50,000 resolution at m/z 200. The AGC targets were set as 2E5 for MS1, 2E4 for MS2 and 5E4 for MS3 with 100 ms, 54 ms and 120 ms ion injections respectively.

## Mass spectrometry data analysis and bioinformatics analysis

The MaxQuant software suite (*Tyanova et al., 2016a*) version 1.6.7.0 was used for database search with the following parameters: reporter ion MS3 type 11plex TMT at a reporter ion mass tolerance of 0.003 Da and the isotopic reporter ion correction factors as provided by the manufacturer were used and searched with an in-built Andromeda search engine, Trypsin/P was selected as a specific enzyme by allowing two missed cleavages, minimum length of 7 amino acids, Oxidation of (M), Acetyl (Protein-N-terminal), deamidation N and Q, phosphorylation of S, T, Y were selected as variable modifications. Carbamidomethylation Cys was selected as a fixed modification. First search tolerance of 20 ppm and main search tolerance of 4.5 ppm were selected. Global Parameters: Uniprot Human protein database (release 2017–02; 42,101 sequences) was used for the database search and 1% protein and peptide false discovery rate was enabled with match between runs enabled within adjacent fractions. For protein quantification, minimum ratio count was set at two for accurate reporter ion quantification. The MaxQuant output protein group text files were processed using Perseus software suite (*Tyanova et al., 2016b*), version 1.6.2.3. The data was filtered for any proteins identified only by site, common contaminants and reverse hits and proteins identified with single unique peptides. The reporter ion intensities were log2 transformed and the data was normalized by subtracting the median for each sample independently. Student-T test and permutation-based false discovery rate of 5% was applied to identify the differentially enriched and significant protein groups between HA-PPM1H[D288A]+LRRK2 and wild-type HA-PPM1H+LRRK2.

## Sample preparation for stoichiometry of Rab10 phosphorylation analysis

50 µg of PPM1H-KO and wild-type A549 cells lysate (lysis buffer containing 50 mM Tris–HCl, pH 7.4, 1% (by vol) Triton X-100, 10% (by vol) glycerol, 0.15 M NaCl, 1 mM sodium orthovanadate, 50 mM NaF, 10 mM 2-glycerophosphate, 5 mM sodium pyrophosphate, 1 µg/ml microcystin-LR, and complete EDTA-free protease inhibitor cocktail) from three independent clones of CRISPR PPM1H-KO and wild-type cells that were treated with 100 nM MLi2 or DMSO and separated on a 4–20% Bis-Tris gel. The gel was fixed with (4:5:1 ratio of water:methanol:acetic acid) for 10 min and then washed with water twice and stained with Colloidal Coomassie stain (NuPAGE Product no: LC6025. Invitrogen). The gel was then washed and de-stained overnight. The In-gel digestion was performed by excising the region between 30 to 20 kDa and de-stained with 40 mM ammonium bicarbonate in 40% (by vol) acetonitrile. Gel pieces were then reduced by adding dithiothreitol to a final

concentration of 5 mM in 100 μl 40 mM ammonium bicarbonate buffer and incubated on a Thermo-mixer with agitation at 1200 rpm for 20 min. The proteins were alkylated by adding 40 mM Iodoace-tamide in 100 μl of 40 mM ammonium bicarbonate buffer and incubated in dark at room temperature. The gel pieces were dehydrated by adding 100 ul of 100% acetonitrile. Trypsin diges-tion was carried out in 0.5% (by mass) sodium deoxycholate with 400 ng trypsin and digested over-night. Peptide extraction was carried out by adding 100 μl of peptide extraction buffer containing 1% (by vol) trifluoroacetic acid in isopropanol and kept under agitation on a Thermomixer at 1200 rpm for 20 min. The supernatant was collected into new collection tubes and peptide extraction was repeated another two times and the peptides were pooled and directly loaded onto SDB-RPS stage-tip clean up was performed. Briefly, Stage-tips are centrifuged at 1500g at Room temperature for 8 minutes. The flow-through fraction was reapplied and centrifuged. The Stage-tips are washed with 100 ul of loading buffer (1% Trifluroacetic acid (by Vol) in Isopropanol) and washed again with 0.2% Trifluroacetic acid (by vol) and finally eluted by adding 30 ul of Elution buffer 1 (1.25% Ammonium hydroxide (by vol )in 50 % Acetonitrile (by vol). The elution was repeated again by adding 30 ul of Elution buffer 2 (1.25% Ammonium hydroxide (by vol )in 60 % Acetonitrile (by vol). Eluted peptides are snap frozen and Vacuum dried.

Immunoprecipitation using the pan-phospho-specific pRab8 antibody was performed as described before (*Lis et al., 2018*) using 2 mg of cell lysate. The immunoprecipitates were washed as described above and elution performed by adding 40 μl of immunoprecipitate (50% slurry) to 40 μl of 4x LDS (106 mM Tris HCl, 141 mM Tris Base, 2% (by mass) LDS, 10% (by vol) glycerol, 0.51 mM EDTA, 0.22 mM SERVA Blue G250, 0.175 mM Phenol Red, pH 8.5). The immunoprecipitated eluates were electrophoresed on 4–20% Bis-Tris gels and the In-gel digestion and SDB-RPS peptide clean-up was performed as explained above. 10% of the immunoprecipitates were retained for immuno-blot analysis.

## LC-MS/MS analysis for stoichiometry of Rab10 phosphorylation analysis

The desalted peptides from Rab10 stoichiometry experiment were dissolved in 50 μl of solvent-A (0.1% (by vol) formic acid in water) and spiked with an equimolar amount of 10 fmol of Rab10 and pRab10 heavy synthetic peptides containing the following sequences (FHpTITTSYYR, FHTITTSYYR, NIDEHANEDVER and AFLTLAEDILR) prior to loading on to Evotips (https://www.evosep.com/) as described in *Bache et al. (2018)*. Briefly, the Evotips are activated by incubating in 100% isopropa-nol and then washed twice with 20 μl solvent-B (100% Acetonitrile in 0.1% formic acid by volume) and subsequently equilibrated by washing twice with 20 μl solvent-A (0.1% (by vol) formic acid in water). The acidified peptide digest was applied onto Evotips and washed twice with 20 μl solvent-A; an additional 100 μl solvent-A was added before placing them onto an auto sampler rack of Evo-Sep-ONE liquid chromatography system connected to a Thermo Q Exactive HFX mass spectrometer (Thermo Fisher Scientific, Bremen, Germany). As described in *Bache et al. (2018)*, the EvoSep One LC system uses a partial elution by applying 35% solvent-B (100% acetonitrile in 0.1% (by vol) formic acid) through the Evotip and subsequently the eluted peptides are pushed into a long storage capil-lary in line with a high pressure pump by a 6-port rotary valve. Following the partial elution, the high-pressure pump pushes the peptide sample into an 8 cm analytical column (PN: EV-1064; 100 um ID, 8 cm long, C18 AQ, 3 μm beads., Dr Maisch, Ammerbuch, Germany) and it is then electro sprayed directly into the mass spectrometer. The peptides were resolved using a 21 min run method and acquired in PRM mode on QE-HFX comprising a full scan covering a range of m/z 300 to 800 and a PRM scan with an inclusion list as provided in a *Supplementary file 2*. The full scan was acquired at 120,000 resolution at m/z 200 using an Ultra-high filed Orbitrap mass analyzer with a 3E6 AGC target. The MS2 spectra in PRM mode was acquired at 30,000 resolution at m/z 200 by tar-geting 1E5 ions with a 300 ms Ion injection times and measured using an Ultra-high filed Orbitrap mass analyzer. The ions were isolated using a Quadrupole mass analyzer with a 0.7 Da isolation win-dow and fragmented using beam-type collisional induced dissociation with a normalized collision energy 27 ev. The loop count was maintained at 10 per duty cycle. The MS1 scans were acquired at 120,000 resolution at m/z 200 and measured in Ultra high-field Orbitrap mass analyzer. The pRab PRM samples were prepared as explained above by spiking an equimolar ratio of 50 fmol heavy pRab1(FRpTITSSYYR), pRab3(YRpTITTAYYR). pRab8(FRpTITTAYYR), pRab10 (FHpTITTSYYR), pRab35(FRpTITSTYYR) and pRab43 (FRpTITQSYYR) along with 25 fmol of Thermo PRTC peptide mix (Product number: 88320. Pierce Thermo Fisher Scientific). The LC-MS/MS analysis of phospho-Rab

PRM samples were analyzed on QE-HFX mass spectrometer interfaced with EvoSep LC -system by selecting 21 min run method described above. The full scan and PRM method were designed as outlined above for full scan and MS2 PRM scans by providing the pRab inclusion list (*Supplementary file 2*). The dynamic retention time was adjusted by enabling on-the-fly retention time correction using PRTC peptide mix to accurately target phospho-Rab peptides. As the six pRabs along with heavy and counterpart endogenous peptides elute between 6–8 min of the LC-gradient, on-the-fly RT correction is necessary to accumulate enough ions and to attain a minimum of 10 data points for accurate peptide identification and quantification. The MS2 spectra AGC target was set at 1E5 and 300 ms Ion injection times were set for the targets.

## Data analysis for stoichiometry of Rab10 phosphorylation analysis

The Rab10 Stoichiometry PPM1H-knock-out (±MLi2) and wild-type (±MLi2) raw mass spectrometry data was imported into Skyline software suite (version 19.1.0.193. MacCoss lab, Dept of Genome Science, University of Washington, USA, *MacLean et al., 2010*). Using heavy labeled synthetic peptides, a PRM spectral library was generated and the peptide precursor and M+1, M+two isotopes and six most abundant fragment ions were selected for the peptide quantification. A heavy to light ratio for each of the peptide was determined. The detail was provided in *Supplementary file 2*. The peptide level Rab10 stoichiometry was determined as explained (*Wu et al., 2011*) in Rab10 stoichiometry: Light phosphopeptide/Heavy phosphopeptide/((Light phosphopeptide/Heavy phosphopeptide)+(Light non-phosphopeptide+Heavy non-Phosphopeptide))X100. GraphPad Prism software version 8.0.1 was used for the downstream analysis.

## Expression and purification of recombinant PPM1H

four × 500 ml of Lysogeny broth containing 100 µg/ml ampicillin was inoculated with a single colony of BL21-CodonPlus(DE3)-RIPL strain of *E. coli* transformed with either Plasmid DU62835 (expresses His-SUMO-PPM1H[wild-type]), Plasmid DU68104 (His-SUMO-PPM1H[H153D]), or Plasmid DU68087 (His-SUMO-PPM1H[D288A]). Bacteria were cultured at 37˚C until OD600 0.4–0.6. The temperature was reduced to 15˚C and protein expression was induced by addition of isopropyl β-D-1-thiogalacto-pyranoside to 50 µM in addition to $MnCl_2$ to 2 mM as PPM family of phosphatases require Mn or Mg for stability (*Das et al., 1996*). Cells were cultured for 16 hr before harvesting by centrifugation at 4200 x g for 20 min at 4˚C. The pellet was resuspended in 200 ml of ice cold *E. coli* lysis buffer (50 mM Tris/HCl pH7.5, 150 mM NaCl, 1% (by vol) Triton, 2 mM $MnCl_2$, 0.5 mM TCEP (tris(2-carbox-yethyl)phosphine)), 1 mM Pefabloc (4-(2-aminoethyl)-benzene-sulfonyl fluoride) and 1 mM benzamidine. Cells were lysed using a cell disruptor (passing sample through twice) and extracts clarified by centrifugation at 30,000 x g for 20 min at 4˚C. Lysates were incubated in 2 ml of Cobalt-Agarose (Amintra Cobalt NTA Affinity Resin, Expedeon) that was equilibrated in *E. coli* lysis buffer and incubated on a roller at 4˚C for 90 min. The resin was loaded onto a column and washed with 20 column volumes of High Salt Wash Buffer (50 mM Tris/HCl pH7.5, 500 mM NaCl, 2 mM MnCl2, 0.03% (by vol) Brij 35, 20 mM Imidazole, 0.5 mM TCEP) until no unbound protein was present in the flow-through. The column was washed with five column volumes of Low Salt Wash buffer (50 mM Tris/HCl pH7.5, 150 mM NaCl, 2 mM MnCl2, 10 mM Imidazole, 0.03% (by vol) Brij 35, 0.5 mM TCEP). Protein was eluted with Elution Buffer (Low Salt Wash + 500 mM imidazole pH7.5) and 1 ml fractions collected. The fractions containing protein were pooled and concentrated to 1 ml final volume using an Amicon Ultra-15 30 kDa concentrator (Millipore, Z717185). This was subjected to gel filtration purification on a Superdex 75 Increase 10/300 GL column equilibrated in Equilibration Buffer (50 mM Tris/HCl pH7.5, 150 mM NaCl, 2 mM MnCl2, 0.5 mM TCEP), collecting 0.2 ml fractions at a flow rate of 0.2 ml/min. Fractions containing wild-type and mutant His-Sumo-PPM1H eluted at the expected molecular weight for a 68 kDa protein were pooled and dialyzed into Equilibration Buffer containing 270 mM Sucrose and stored in aliquots at −80˚C.

## Expression and purification of Rab8A

Rab8A (residues 1–181) was produced in six versions: the Rab8A [Q67L] mutant in the presence of 1 µM GTP-gamma-S and the wild-type in the presence of 10 µM GDP or in the presence of 5 µM GTP-gamma-S. All three were made as unphosphorylated and phosphorylated proteins.

Rab8A 1–181 [Q67L] was produced in GTP-gamma-S as follows. The codon optimized coding sequence of Rab8A residues 1–181 [Q67L] subcloned into pET28b was kindly provided by Amir Khan (Trinity College Dublin) and expressed based on a protocol provided by the Khan lab. Briefly, the expression construct was transformed into BL21 DE3 and expressed in Lucia Broth medium (Merck, Darmstadt) supplemented with 50 μg/ml Kanamycin (Formedium, UK). Typically, 12 × 1L expressions were set up. When the optical density at 600 nm reached 0.6, the culture temperature was dropped to 18°C and after 60 min, Rab8A expression was induced by addition of Isopropyl β-D-1-thiogalactopyranoside (Formedium, UK) to 0.1 mM for incubation overnight. The cells were collected by sedimentation and resuspended in ice cold lysis buffer (50 mM Tris pH 7.5, 10% (by vol) glycerol, 250 mM NaCl, 15 mM Imidazole, 2 mM MgCl$_2$, 0.4% (by vol) Triton X-100, 1 mM Pefabloc, 10 μg/ml Leupeptin) employing 20 ml Lysis buffer per 5 ml of cell sediment. The cell suspension was sonicated, and insoluble material was removed by centrifugation (35000 x g at 4°C for 30 min). The supernatant was incubated with 3 ml Ni-agarose (Expedion, UK) for 90 min, which was then washed 5 times with 10 vol wash buffer (50 mM Tris pH 7.5, 5% (by vol) glycerol, 250 mM NaCl, 15 mM Imidazole, 14 mM 2-mercaptoethanol, 2 mM MgCl$_2$, 0.03% (by vol) Brij35, 1 μM GTP-gamma-S). The protein was eluted with 4 × 1 resin volume of elution buffer (30 mM Tris pH 7.5, 3% (by vol) glycerol, 0.4 M imidazole, 8 mM 2-mercaptoethanol, 1.2 mM MgCl$_2$, 0.02% (by vol) Brij35, 0.6 μM GTP-gamma-S) and dialyzed against 50 mM Tris pH 7.5, 10% (by vol) glycerol, 250 mM NaCl, 14 mM 2-mercaptoethanol, 2 mM MgCl$_2$, 0.6 μM GTP-gamma-S in the presence of 60 Units of Thrombin (Sigma-Aldrich T1063-1KU). The sample was then passed through a Ni-agarose column (3 ml) equilibrated in 50 mM Tris pH 7.5, 10% (by vol) glycerol, 250 mM NaCl, 14 mM 2-mercaptoethanol, 2 mM MgCl$_2$ and 0.6 μM GTP-gamma-S. The protein that was eluted (typically 25 mg) was subjected to chromatography on a Superdex-75 XK 26/60 column equilibrated in 50 mM Tris pH 7.5, 10% (by vol) glycerol, 250 mM NaCl, 14 mM 2-mercaptoethanol, 2 mM MgCl$_2$ and 0.6 μM GTP-gamma-S, collecting 1.5 ml fractions at a flow rate of 2.5 ml/min. Fractions containing Rab8A that eluted at the expected molecular weight region for a 20 kDa protein, were either concentrated to >1 mg/ml to be used as unphosphorylated Rab8A or - for the production of phospho-Rab8A - directly phosphorylated, avoiding both concentration and freeze thawing to mitigate aggregation. Up to 16 mg of Rab8A could be produced from a 12 Liter preparation.

Wild-type Rab8A 1–181 (wild-type, truncated) was produced either in the presence of 10 μM GDP or in the presence of 5 μM GTP-gamma-S respectively present in all solutions throughout. Otherwise the expression and purification method were identical to the method used for the purification of Rab8A 1–181 [Q67L].

## Expression and purification of Rab10

The method for expressing and purifying Rab10[1-181] (GDP-bound) was similar to the method for purifying GDP-bound Rab8A. Rab10 is much more aggregation prone than Rab8A and we found that the addition of 20 mM L-arginine and 0.5 M NaCl in the lysis buffer, wash buffers and dialysis buffer was useful to mitigate aggregation. Yield was over 10 times lower than that obtained for Rab8A.

## Generation of Thr72 stoichiometrically phosphorylated Rab8A (pRab8A)

MST3 kinase can specifically and efficiently phosphorylate Rab8A at Thr72 in vitro (*Vieweg et al., 2019*). As MST3 is much easier to express than LRRK2, we decided to phosphorylate Rab8A at Thr72 using recombinant MST3. 6His-MST3 residues 1–431 (DU62878) was obtained from MRC-PPU Reagents and Services (https://mrcppureagents.dundee.ac.uk/reagents-proteins/overview). 5 mg of His-MST3 were incubated with 12 mg Rab8A in a total volume of 20 ml in Equilibration buffer, supplemented with 10 mM MgCl$_2$ and 1 mM ATP at 27°C overnight. 6His-MST3 was removed from the reaction by passing the reaction through a 3 ml Ni-agarose column equilibrated in 50 mM Tris pH 7.5, 5% (by vol) glycerol, 250 mM NaCl, 14 mM 2-mercaptoethanol, 15 mM Imidazole, 2 mM MgCl2, 0.03% (by vol) Brij35, 1 μM GTP-gamma-S). The phosphorylated Rab8A sample was diluted 7-fold into Source-S Buffer [20 mM MES pH 5.3, 10 mM NaCl, 10% (by vol) glycerol, 0.03% (by vol) Brij 35, 14 mM 2-mercaptoethanol, 2 mM MgCl$_2$, 1 μM GTP gamma-S. The phosphorylated Rab8A was separated from the unphosphorylated Rab8A by chromatography over a Source 15 S HR10/10 column.

The column was equilibrated and run in Source-S Buffer at a flowrate of 2 ml/min and a gradient from 10 mM NaCl to 0.5 M NaCl deployed and 1 ml fractions were collected. Phosphorylated Rab8A eluted at around 10 mS/cm (~80–110 mM NaCl), whereas a much smaller amount of non-phosphorylated Rab8A eluted after 15 mS/cm (*Figure 8—figure supplement 1*). Analysis of the phosphorylated and unphosphorylated Rab8A on 12% Phos-tag gels (*Figure 8—figure supplement 1B*) and mass spectrometry (*Figure 8—figure supplement 2*) confirmed that the first eluting peak was only significantly phosphorylated at Thr72 and the stoichiometry of phosphorylation was judged to be ~97%. The fractions containing the phosphorylated Rab8A were concentrated, pooled, aliquoted, snap frozen in liquid nitrogen and stored at −80℃. GDP bound Rab8A was produced by supplementing all solutions during the preparation with 10 µM GDP/10 mM MgCl$_2$. Likewise, GTPgammaS Rab8A was produced in the presence of 2–5 µM GTPgammaS/10 mM MgCl$_2$ throughout the preparation.

## Generation of phosphorylated Rab10 protein

2.2 mg of monomeric, pure Rab10 residues 1–181 was incubated with 0.5 mg His-MST3 in the presence of 10 mM MgCl2 and 1 mM ATP at 27℃ for 16 hr. The kinase was removed by depletion over a 0.2 ml Ni-agarose bed. The flow-through was diluted 15-fold into 20 mM MES pH 5.3, 10 mM NaCl, 10% (by vol) glycerol, 0.03% (by vol) Brij 35, 14 mM 2-mercaptoethanol, 2 mM MgCl$_2$, 10 µM GDP and purified over a 10 ml Source 15 s column as described for Rab8A [Q67L]. The column was developed with a NaCl gradient, here in the presence of 10 µM GDP. Recovery was poor, but 0.1 mg of phospho Rab10 could be obtained. Phospho Rab10 eluted at 14 mS/cm and unphosphorylated Rab10 at 15 and 17 mS/cm respectively (*Figure 8—figure supplement 3*).

## LC-MS/MS and stoichiometry analysis of phosphorylated Thr72 Rab8A and phosphorylated Thr73 Rab10

2 ug of Rab8A 1–181 [Q67L] and phospho-Rab8A 1–181 [Q67L] and 2 ug of wild-type Rab10 and wild-type phospho-Rab10 in technical triplicates was resolved on a 4–12% Bis-Tris gradient gel. Following colloidal Coomassie staining and destaining, the lane encompassing 25 kDa region was excised and subjected to In-gel digestion using trypsin and quenched the reaction by adding final 1% trifluoroacetic acid (by vol) as explained above and desalted using C18 stage-tips. Briefly, A single 16 guage C18 disk was prepared in-house into a 250 ul pipette tip. The C18 stage-tips were activated by adding 60 ul of 100% Acetonitrile and subsequently equilibrated twice with solvent-A (0.1% trifluoroacetic acid by Vol). The acidified peptide digest was loaded onto the Stage-tips and after centrifugation the flow through was reapplied and then washed twice with 60 ul of Solvent-A. The desalted peptides were eluted by adding 30 ul of solvent-B (50% acetonitrile in 0.1% trifluoroacetic acid (by vol) and immediately snap frozen and vacuum dried in a Speed-Vac. The peptide samples were subjected to LC-MS/MS analysis by preparing the EvoTips as explained above and analyzed on QE-HFX mass spectrometer by selecting 44 min run method in a data dependent mode. MS1 precursor ions in the range of m/z 350 to 1500 was isolated and analyzed using Ultra high-filed Orbitrap mass analyzer at a resolution of 120,000 at m/z 200. Top 10 precursor ions were selected for the MS2 by isolating using quadrupole mass analyzer with 0.7 Da isolation width and fragmented using beam type collional induced dissociation (HCD) with a normalized collision energy 27 ev and measured using Ultra high-filed Orbitrap mass analyzer at a resolution of 30,000 at m/z 200.

The raw mass spectrometry data was processed using MaxQuant version 1.6.6.0 as explained above by enabling Label free quantification. The Rab8A 1–181 [Q67L] Phosphopeptide FRpTITTAYR, unmodified peptide TITAYR intensities and pRab8A 1–181 [Q67L] and Rab8A 1–181 [Q67L] protein LFQ intensities were used to determine the FRpTITTAYR, phosphorylation stoichiometry. To enable accurate phosphosite stoichiometry to be calculated both the modified and unmodified Thr72 peptides were excluded in determining the protein LFQ intensities. Similar analysis was done for the Rab10 Thr73 Phosphopeptide FHpTITTSYYR, unmodified Thr 73peptide FHTITTSYYR intensities and Rab10 [Wt] and pRab10 [Wt] protein LFQ intensities were used to determine the FHpTITTSYYR, phosphorylation stoichiometry. To enable accurate phosphosite stoichiometry to be calculated both the modified and unmodified Thr73 peptides were excluded in determining the protein LFQ intensities.

## PPM1H phosphatase assays employing phospho-Rab8A as substrate

In vitro dephosphorylation assay was performed in a total volume of 20 µl in 40 mM Hepes buffer (pH 7.0) containing 10 mM $MgCl_2$ using 2.5 µg pT72 phosphorylated Rab8a[Q67L] and varying levels of recombinant PPM1H. 3 µl of pT72 phosphorylated Rab8a[Q67L] at 0.83 mg/ml in a buffer containing 20 mM MES pH 5.3, 0.1 M NaCl, 10% (by vol) glycerol, 0.03% (by vol) Brij 35, 14 mM 2-mercaptoethanol, 2 mM $MgCl_2$, 1 µM GTP-γ-S added to the assay. The assay was initiated by addition of 2 µl of serial dilutions of PPM1H (0.1 mg/mL, 0.02 mg/mL, 0.004 mg/mL, 0.0008 mg/mL) diluted into HEPES Buffer from a stock of 1.73 mg/ml PPM1H in 50 mM Tris/HCl pH7.5, 150 mM NaCl, 2 mM MnCl2, 0.5 mM TCEP Buffer. The assay was carried out for 30 min and terminated by addition of 6 µl 4 x LDS (106 mM Tris HCl, 141 mM Tris Base, 2% (by mass) LDS, 10% (by vol) glycerol, 0.51 mM EDTA, 0.22 mM SERVA Blue G250, 0.175 mM Phenol Red, pH 8.5) with 5% (by vol) 2-mercaptoethanol. Samples were then subjected to Phos-tag gel electrophoresis to determine stoichiometry of phosphorylated Rab8a as described previously in *Ito et al. (2016)*. Gel was stained using Instant Blue Coomassie (Expedeon).

## Overexpression of GFP-Rab10 and PPM1H in MEF cells

MEF cells stably expressing GFP-Rab10 were transiently transfected with plasmids carrying wild-type HA-PPM1H, HA-PPM1H[H153D], or HA-PPM1H[D288A]. After 24 h the cells were fixed with 4% (by vol) paraformaldehyde for 10 min, permeabilized with 0.1% saponin for 15 min, and blocked with 1% (by mass) BSA for 1 h. Cells were subsequently stained with mouse anti-HA antibody 1:1000 (Sigma-Aldrich H3663) and rabbit phospho-Rab10 1:1000 (Abcam ab230261). Highly cross absorbed H+L secondary antibodies (Life Technologies) conjugated to Alexa 568 or Alexa 647 were used at 1:5000. Primary and secondary antibody incubations were for 1 h at room temperature. All images were obtained using a spinning disk confocal microscope (Yokogawa) with an electron multiplying charge coupled device (EMCCD) camera (Andor, UK) and a 20x1.4NA or 40x1.4NA objective. Images were analyzed using CellProfiler and presented as maximum intensity projections. Results were quantified by determining the ratios of phospho-Rab10 signal to GFP-Rab10 in cells expressing wild-type HA-PPM1H, HA-PPM1H[H153D], or HA-PPM1H[D288A], and normalizing these numbers to the phospho-Rab10/GFP-Rab10 ratio in non-expressing cells. For each condition at least 20 cells were analyzed. Significance was determined by one-way analysis of variance with Dunnett's post-test at 95% confidence interval. ***, $P < 0.001$.

## Immunofluorescence microscopy

RPE cells were transiently transfected with HA-PPM1H plasmid. After 24 h, cells were treated for 2 h with 20 µM nocodazole in DMSO or DMSO alone, fixed with 4% (by vol) paraformaldehyde for 10 min, permeabilized with 0.1% by vol Triton X-100 for 5 min, and blocked with 1% (by mass) BSA for 1 h. Cells were subsequently stained with mouse or rabbit anti-HA antibody (Sigma-Aldrich H3663 or H6908, 1:1000) and the following Golgi markers: rabbit anti-GCC185 1:1000 (serum), rabbit anti-ß-Cop 1:1000 (serum), mouse anti-p115 1:1000 (ascites), mouse anti-cation independent mannose 6-phosphate receptor 1:1 (2G11 culture sup), and rabbit anti-ACBD3 1:1000 (Sigma-Aldrich HPA015594). Highly cross absorbed H+L secondary antibodies (Life Technologies) conjugated to Alexa 488 or Alexa 568 were used at 1:5000. Primary and secondary antibody incubations were for 1 h at room temperature. Nuclei were stained with 0.1 µg/ml DAPI (Sigma). All images were obtained using a spinning disk confocal microscope (Yokogawa) with an electron multiplying charge coupled device (EMCCD) camera (Andor, UK) and a 100x1.4NA oil immersion objective. Images were analyzed using CellProfiler software (*Carpenter et al., 2006*) and presented as maximum intensity projections.

## Ciliation

MEF cells were infected in two rounds (day 1 and day 3) with lentiviruses encoding the shRNA sequences and on day 5, infected cells were selected using puromycin for 72 hr as described (*Dhekne et al., 2018*); pools of stably infected cells were then assayed for cilia formation. Ciliation was monitored after 24 hr serum starvation using anti-Arl13B antibody (NeuroMab, Davis, California) to stain cilia for immunofluorescence microscopy (*Dhekne et al., 2018*).

## shRNA sequences

Scrambled
 CCTAAGGTTAAGTCGCCCTCGCTCGAGCGAGGGCGACTTAACCTTAGG

PPM1H shRNA 1
 CCGGGCTACATGAAACCCTATTTAACTCGAGTTAAATAGGGTTTCATGTAGCTTTTTG

PPM1H shRNA 2
 CCGGGCCCGAGTAATGGCAACTATTCTCGAGAATAGTTGCCATTACTCGGGCTTTTTG

PPM1H shRNA 3
 CCGGCCTCACAGGTTTGTGCCTTTACTCGAGTAAAGGCACAAACCTGTGAGGTTTTTG

PPM1H mRNA was monitored by qPCR as described (*Dhekne et al., 2018*). Briefly, cells were lysed in Trizol (Sigma) for 5 min. RNA extraction was performed using chloroform/isopropyl alcohol; cDNA was synthesized from 600 ng RNA using a high capacity cDNA synthesis kit with multiscribe reverse transcriptase (Applied Biosystems). The cDNA was diluted 100-fold and 1 µl was used as template in the PowerUp SYBR green 2X master mix and analyzed in a qPCR machine (7900HT Applied Biosystems). Gene expression was compared using the 2(delta-delta Ct) method with GAPDH as internal control. The primers used were: PPM1H primer Set 1 (5′−3′), forward: ATA TGGAGAAGGCAAGAAGG, reverse: TCATATCTGGAGAGATCGTAG and PPM1H primer Set 2 (5′−3′) forward: AAAACCATTCCTGTCTTCAG reverse: TCTGGATCACAGTTAGGAAG.

## Acknowledgements

We thank Amir Khan for providing us with the codon optimized Rab8a[1-181, 67QL], Ilaria Volpi for helping assembly of the phosphatase siRNA library, Miratul Muqit for providing the rabbit polyclonal Rab8A phospho-S111 antibody, Nicole K Polinski, Marco Baptista and Shalini Padmanabhan (Michael J Fox Foundation for Parkinson's research) for helpful discussions, the excellent technical support of the MRC-Protein Phosphorylation and Ubiquitylation Unit (PPU) DNA Sequencing Service (coordinated by Gary Hunter), the MRC-PPU tissue culture team (coordinated by Edwin Allen), MRC PPU Reagents and Services antibody and protein purification teams (coordinated by Hilary McLauchlan and James Hastie). We also thank Dr. Renata F.Soares from MRC-PPU mass spectrometry facility with her help in the maintenance of the LC and mass spectrometer for the analysis. This work was supported by the Michael J Fox Foundation for Parkinson's research [grant number 17298 (to SRP and DRA)] and [grant number 6986 (to SRP and DRA)] and MJFF Langston Award (to DRA which was used to purchase the siRNA library); the Medical Research Council [grant number MC_UU_12016/2 (to DRA)]; the pharmaceutical companies supporting the Division of Signal Transduction Therapy Unit (Boehringer-Ingelheim, GlaxoSmithKline, Merck KGaA -to DRA) and the U.S. National Institutes of Health DK37332 (to SRP).

## Additional information

### Funding

| Funder | Grant reference number | Author |
|---|---|---|
| Michael J. Fox Foundation for Parkinson's Research | 17298 | Suzanne R Pfeffer<br>Dario R Alessi |
| Michael J. Fox Foundation for Parkinson's Research | 6986 | Suzanne R Pfeffer<br>Dario R Alessi |
| Michael J. Fox Foundation for Parkinson's Research | Langston Award | Dario R Alessi |
| Medical Research Council | MC_UU_12016/2 | Dario R Alessi |
| National Institutes of Health | DK37332 | Suzanne R Pfeffer |

The funders had no role in study design, data collection and interpretation, or the decision to submit the work for publication.

## Author contributions

Kerryn Berndsen, Conceptualization, Data curation, Validation, Investigation, Visualization, Methodology, Designed and executed experiments in Figures 2, 3, 5, 6, 7, 8, 9 with PL and was involved in discussing and interpreting the data; Pawel Lis, Conceptualization, Data curation, Formal analysis, Supervision, Investigation, Visualization, Methodology, Writing—original draft, Writing—review and editing, Designed and executed experiments in Figures 2, 3, 5, 6, 7, 8, 9 with KB and was involved in discussing and interpreting the data; Wondwossen M Yeshaw, Data curation, Formal analysis, Validation, Investigation, Visualization, Designed and executed Figure 11, Designed and executed Figures 4 and 10 with PSW and was involved in discussing and interpreting the data; Paulina S Wawro, Data curation, Investigation, Visualization, Methodology, Designed and executed Figures 4 and 10 with WMY and was involved in discussing and interpreting the data; Raja S Nirujogi, Data curation, Formal analysis, Investigation, Visualization, Designed and executed mass spectrometry experiments for Figures 6B, 6D, 9B, Figure 8—Figure supplement 2, Figure supplement 5 and expression analysis of PPM1H and PPM1M in Figure 12—Figure Supplements 1 and 2 and was involved in discussing and interpreting the data; Melanie Wightman, Data curation, Investigation, Methodology, Undertook most of the cloning required for this study; Thomas Macartney, Data curation, Investigation, Methodology, Generated expression constructs for CRISPR/CAS9 gene editing studies; Mark Dorward, Investigation, Methodology, Developed the expression and purification system to produce active recombinant PPM1H, PPM1M and PPM1J phosphatases, and MST3 kinase; Axel Knebel, Conceptualization, Formal analysis, Validation, Investigation, Visualization, Methodology, Expressed, purified and phosphorylated Rab8A for experiments shown in Figure 8, Involved in discussing and interpreting the data; Francesca Tonelli, Conceptualization, Data curation, Formal analysis, Investigation, Visualization, Designed and executed experiments in Figure 1, Involved in discussing and interpreting the data; Suzanne R Pfeffer, Conceptualization, Data curation, Supervision, Funding acquisition, Visualization, Writing—original draft, Project administration, Writing—review and editing, Supervised the project with DRA and wrote the manuscript; Dario R Alessi, Conceptualization, Formal analysis, Supervision, Funding acquisition, Writing—original draft, Project administration, Writing—review and editing, Supervised the project with SRP and wrote the manuscript

## Author ORCIDs

Pawel Lis (iD) https://orcid.org/0000-0002-4978-7671
Wondwossen M Yeshaw (iD) https://orcid.org/0000-0002-3134-3458
Suzanne R Pfeffer (iD) https://orcid.org/0000-0002-6462-984X
Dario R Alessi (iD) https://orcid.org/0000-0002-2140-9185

## Decision letter and Author response

Decision letter https://doi.org/10.7554/eLife.50416.038
Author response https://doi.org/10.7554/eLife.50416.039

## Additional files

### Supplementary files

• Supplementary file 1. Numerical data of the pRab10/Total Rab10 ratios relative to scrambled siRNA control from Screens 1, 2 and 3 (experiments shown in *Figure 2*). The list of all siRNA targets and the sequences of all siRNA oligonucleotides are also included.
DOI: https://doi.org/10.7554/eLife.50416.031

• Supplementary file 2. The raw data obtained for Rab10 phosphorylation stoichiometry assessment (*Figure 6B*) and Rab phosphorylation levels in the PRM scan (*Figure 6D*).
DOI: https://doi.org/10.7554/eLife.50416.032

• Supplementary file 3. The list of protein groups identified in the PPM1H substrate trapping mutant experiment (*Figure 9B*).
DOI: https://doi.org/10.7554/eLife.50416.033

• Transparent reporting form DOI: https://doi.org/10.7554/eLife.50416.034

## Data availability

We have deposited the mass spectrometry raw data and MaxQuant search output tables to ProteomeXchange, PRIDE database https://www.ebi.ac.uk/pride/archive/ unique identifier, PXD014794. We include the primary data for each of the 3 siRNA screens (Figure 2-figure Supplements 1, 2 and 3), in addition to the quantitation of the pRab10/Total Rab10 ratios in Supplementary file 1. The file also contains all RNA sequences of siRNA library. All Plasmids, antibodies and proteins (including datasheets and sequence information) that we have generated for this study can be requested and information downloaded from MRC PPU Reagents and Services (https://mrcppureagents.dundee.ac.uk/).

The following dataset was generated:

| Author(s) | Year | Dataset title | Dataset URL | Database and Identifier |
|---|---|---|---|---|
| Kerryn Berndsen, Pawel Lis, Raja S Nirujogi | 2019 | PPM1H phosphatase counteracts LRRK2 signaling by selectively dephosphorylating Rab proteins | http://proteomecentral. proteomexchange.org/ cgi/GetDataset?ID= PXD014794 | ProteomeXchange, PXD014794 |

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
