## [Decision Letter]

**Acceptance summary:**

LRRK2 is a protein kinase gene frequently mutated in Parkinson's disease, but precisely how LRRK2 mutations promote Parkinson's Disease is unknown. Some mutations in LRRK2 promote phosphorylation of the Switch II region of several Rab proteins, including Rab10, but how hyper-phosphorylated Rabs are returned to the unphosphorylated state is unknown. Through a focused genetic screen, this paper identifies PPM1H as a phosphatase that can remove phosphate from a subset of Rabs that are targeted by LRRK2. Regulation of Rab phosphorylation status by PPM1H may have implications for trafficking pathways altered by mutant LRRK2. Interestingly, detailed analysis suggests that while PPM1H is a major phosphatase for these Rabs, there may be other redundant phosphatases for one or more of the Rab proteins. This study has major implications for understanding the regulation of Rab phosphorylation and may also suggest ways to reduce the levels of specific phospho-Rab proteins in cells with mutant LRRK2, for example, by elevating the activity of PPM1H.

**Decision letter after peer review:**

Thank you for submitting your article "PPM1H phosphatase counteracts LRRK2 signaling by selectively dephosphorylating Rab proteins" for consideration by *eLife*. Your article has been reviewed by three peer reviewers, including Wade Harper as the Reviewing Editor and Reviewer #1, and the evaluation has been overseen by Philip Cole as the Senior Editor. The following individuals involved in review of your submission have agreed to reveal their identity: Alexandra C Newton (Reviewer #2); Peter S McPherson (Reviewer #3).

The reviewers have discussed the reviews with one another and the Reviewing Editor has drafted this decision to help you prepare a revised submission.

Summary:

This paper identifies PPM1H as a phosphatase for various Rabs that are phosphorylated on their Switch II region by LRRK2. Noting that Rab phosphorylation is rapidly reversed (within a couple of min) upon inhibition of LRRK2 by a small molecule inhibitor, the authors first demonstrated that PP1 or PP2A are not required for Rab dephosphorylation. They subsequently performed a focused siRNA screen (264 phosphatases) to identify the Rab phosphatase. From 3 screens, PPM1H, a metal dependent phosphatase emerged as the lead candidate. The authors performed a number of secondary assays that indicate that PPM1H is the primary Rab phosphatase for the LRRK2 phospho-site. Closely related family members J and M have very weak activity in vitro. The M isoform appeared to have some activity when overexpressed in cells but it also seems to slightly reduce LRRK2 levels. Using immunofluorescence, the authors demonstrated that PPM1H associates with cis and medial Golgi, as well as being localized to the cytosol. Overall, the reviews were very positive.

After discussion among the reviewer's, we suggest the following revisions:

Essential revisions:

It remains somewhat unclear as to whether or not all of the Rab substrates of LRRK2 (3A-D, 8, 10, 12, 29, 35, 43) are physiologically relevant targets of PPM1H. The reviewer's feel it would be useful to examine the relevance of PPM1H on additional LRRK2-targeted Rabs, as performed in Figure 4. Of course this experiment will require appropriate high-quality phosphospecific antibodies, and these reagents may limit the number of Rabs that could be tested in this way. One could also envision couple a phos-tag gel assay to overexpression of WT and mutant (substrate trapping) PPM1H, were suitable antibodies not available for any of the additional p-Rabs. Expansion of the PPM1H substrate repertoire for any additional LRRK2-targets Rabs would strengthen the idea that the phosphatase is acting broadly in opposition to LRRK2.

---

## [Author Response]

Essential revisions:It remains somewhat unclear as to whether or not all of the Rab substrates of LRRK2 (3A-D, 8, 10, 12, 29, 35, 43) are physiologically relevant targets of PPM1H. The reviewer's feel it would be useful to examine the relevance of PPM1H on additional LRRK2-targeted Rabs, as performed in Figure 4. Of course this experiment will require appropriate high-quality phosphospecific antibodies, and these reagents may limit the number of Rabs that could be tested in this way. One could also envision couple a phos-tag gel assay to overexpression of WT and mutant (substrate trapping) PPM1H, were suitable antibodies not available for any of the additional p-Rabs. Expansion of the PPM1H substrate repertoire for any additional LRRK2-targets Rabs would strengthen the idea that the phosphatase is acting broadly in opposition to LRRK2.

Thank you-this is an important question. In order to assess the effect of PPM1H knock-out on the phosphorylation of additional endogenous Rab proteins in the A549 cells, we undertook an immunoprecipitation experiment using a pan- phospho-specific-Rab antibody that cross-reacts with LRRK2 phosphorylated Rab3, Rab8, Rab10, Rab35 and Rab43 (PMID 29127256). This approach has previously been exploited to demonstrate that this set of Rab proteins are phosphorylated by LRRK2 in vivo (Lis et al., 2018). Immunoblot analysis of the pan-phospho-specific-Rab immunoprecipitates revealed that in addition to Rab10, phosphorylated forms of Rab8A and to a lower extent Rab35 were enriched in the PPM1H knock-out cell lines in comparison to wild-type (revised Figure 6C). We were unable to detect Rab3A or Rab43 in these studies, suggesting that the levels of the LRRK2 phosphorylated forms of these substrates are perhaps too low to easily detect in A549 cells. We also immunoblotted cell extracts with a phospho-antibody that specifically detects LRRK2 phosphorylated Rab12, revealing that LRRK2 phosphorylated Rab12 levels were similar in wild type and PPM1H knock-out cell extracts (revised Figure 6C). This study suggests that Rab12 phosphorylation may not be regulated by PPM1H, at least in A549 cells.

In addition, we subjected the pan-specific phospho-Rab immunoprecipitates to mass spectrometry analysis that confirmed significantly higher levels of Rab8 and Rab10 and to a lower extent Rab35 in samples immunoprecipitated from wild type compared to PPM1H knock-out cells (revised Figure 6D). We weakly detected Rab3A but its levels were very similar between wild type and PPM1H knock-out cells.

We have also expanded our analysis by which Rab proteins are dephosphorylated by LRRK2 in an overexpression analysis. We had previously shown that overexpressed PPM1H can dephosphorylate LRRK2 phosphorylated Rab3A, Rab8A, Rab10, Rab12, Rab35 and Rab43 but have now extended this panel to show that Rab3B, Rab3C, Rab3D and Rab8B are also dephosphorylated by overexpressed PPM1H (revised Figure 3B, Figure 3C and Figure 3—figure supplement 1A to 1H).